# Unraveling the roles of coastal bacterial consortia in degradation of various lignocellulosic substrates

Qiannan Peng,[1] Lu Lin,[1] Qichao Tu,[1] Xiaopeng Wang,[2] Yueyue Zhou,[2] Jiyu Chen,[1] Nianzhi Jiao,[3,4] Jizhong Zhou[5,6,7,8]

**ABSTRACT**  Lignocellulose, as the most abundant natural organic carbon on earth, plays a key role in regulating the global carbon cycle, but there have been only few studies in marine ecosystems. Little information is available about the extant lignin-degrading bacteria in coastal wetlands, limiting our understanding of their ecological roles and traits in lignocellulose degradation. We utilized *in situ* lignocellulose enrichment experiments coupled with 16S rRNA amplicon and shotgun metagenomics sequencing to identify and characterize bacterial consortia attributed to different lignin/lignocellulosic substrates in the southern-east intertidal zone of East China Sea. We found the consortia enriched on woody lignocellulose showed higher diversity than those on herbaceous substrate. This also revealed substrate-dependent taxonomic groups. A time-dissimilarity pattern with increased alpha diversity over time was observed. Additionally, this study identified a comprehensive set of genes associated with lignin degradation potential, containing 23 gene families involved in lignin depolymerization, and 371 gene families involved in aerobic/anaerobic lignin-derived aromatic compound pathways, challenging the traditional view of lignin recalcitrance within marine ecosystems. In contrast to similar cellulase genes among the lignocellulose substrates, significantly different ligninolytic gene groups were observed between consortia under woody and herbaceous substrates. Importantly, we not only observed synergistic degradation of lignin and hemi-/cellulose, but also pinpointed the potential biological actors at the levels of taxa and functional genes, which indicated that the alternation of aerobic and anaerobic catabolism could facilitate lignocellulose degradation. Our study advances the understanding of coastal bacterial community assembly and metabolic potential for lignocellulose substrates.

**IMPORTANCE**  It is essential for the global carbon cycle that microorganisms drive lignocellulose transformation, due to its high abundance. Previous studies were primarily constrained to terrestrial ecosystems, with limited information about the role of microbes in marine ecosystems. Through *in situ* lignocellulose enrichment experiment coupled with high-throughput sequencing, this study demonstrated different impacts that substrates and exposure times had on long-term bacterial community assembly and pinpointed comprehensive, yet versatile, potential decomposers at the levels of taxa and functional genes in response to different lignocellulose substrates. Moreover, the links between ligninolytic functional traits and taxonomic groups of substrate-specific populations were revealed. It showed that the synergistic effect of lignin and hemi-/cellulose degradation could enhance lignocellulose degradation under alternation of aerobic and anaerobic conditions. This study provides valuable taxonomic and genomic insights into coastal bacterial consortia for lignocellulose degradation.

**KEYWORDS**  lignocellulose degradation, bacterial communities, coastal intertidal zone, priming effect, lignin degradation, functional gene

Address correspondence to Lu Lin, linlu2019@sdu.edu.cn.

The authors declare no conflict of interest.

See the funding table on p. 20.

Lignocellulose, which is mainly composed of lignin, cellulose, and hemicellulose, is the most abundant natural organic carbon biomass on earth, and thus its decomposition is important in regulating the global carbon cycle (1, 2). Microorganisms, as decomposers, can degrade organic matter and drive biogeochemical element cycles (3, 4). Because lignin has long been considered as the most recalcitrant terrestrial organic carbon (TerrOC), knowledge of lignin degrading microorganisms is limited (5, 6), although numerous studies have reported hemi-/cellulose degrading microorganisms (7–9). The well-characterized lignin decomposers are white-rot fungi and brown-rot fungi, e.g., *Basidiomycotina* and *Agaricomycota* (10). White-rot fungi can secrete extracellular peroxidases that attack a variety of C–C bonds in lignin, whereas brown-rot fungi mostly employ extracellular nonenzymatic hydroxyl radicals to break lignin (11). Such rapid delignification allows fungi to access more readily degradable carbohydrate polymers as carbon and energy sources. As a result, they commonly do not catabolize lignin-derived aromatic compounds. Furthermore, lignin-degrading fungi generally prefer moderate environmental conditions, e.g., aerobic zone and moderate temperature, pH, and salinity (5, 12–14). Recently, lignin-degrading bacteria in terrestrial ecosystems (e.g., forest soil, wood, and gut) have increasingly emerged, primarily belonging to three classes: *Actinomycetes* (e.g., *Streptomyces* and *Rhodococcus*), (a-, β-, γ-) *Proteobacteria* (e.g., *Sphingomonas*, *Pandoraea,* and *Pseudomonas*), and *Bacilli* (e.g., *Bacillus*) (15–17). Besides lignin, they can also catabolize various lignin-derived aromatic compounds. Moreover, these bacteria display a broad environmental adaptability, such as aerobic/anaerobic, thermophilic, halophilic, and alkalophilic conditions (18–21). Such advantages demonstrate the importance of characterizing bacterial communities to better understand the degradation of lignin and lignocellulosic biomass.

In contrast to the substantial studies on terrestrial ecosystems (2, 22, 23), only a limited number of bacterial lignin-degradation studies have targeted aquatic ecosystems, especially marine ecosystems (13, 24). This is because that lignin, as a TerrOC, has traditionally been considered to be highly stable and, hence, widely used as a biomarker in aquatic ecosystems, e.g., estuarine and marine ecosystems (25, 26). It is worth noting that while ~0.3 Pg TerrOC is discharged annually to the global ocean, only 0.1 Pg (~30%) TerrOC is buried in marine sediment (27). This suggests that a substantial portion of the "missing" TerrOC in marine systems might be more actively consumed by microorganisms than previously thought. Coastal intertidal zones are located at the land-sea interface, and are characterized by strong dynamic interactions between tides and river runoff, as well as between surface water and groundwater. They act as hotspots of biogeochemical cycling, in which bacteria play a critical role (28–30). Previous studies suggested that communities by short-term (days) selection are firstly shaped by stochastic processes with the inherent randomness, followed by deterministic processes to select for or against specific microorganisms (31–33). Substrates, as available nutrients, should be the strong selection pressure for community assembly. Meanwhile, exposure time is a comprehensive factor, and may include multiple environmental factors (e.g., temperature, pH, salinity, and dissolved oxygen [DO]). However, our understanding about coastal bacterial community assembly under long-term selection (months) remains surprisingly poor. Therefore, it is important to explore the bacterial community assembly and composition with lignocellulosic metabolic potential in coastal ecosystems, which can then be incorporated into global carbon models.

Lignin is a complex heterogeneous phenylpropanoid polymer. It is composed of guaiacyl (G) units, syringyl (S) units, and p-hydroxyphenyl (H) units, linked by a variety of ether and C–C bonds (34). Moreover, lignin content and composition can vary substantially from species to species, e.g., herbaceous, hardwood, and softwood (35). Such complex and dynamic structures are in sharp contrast to extant studies which utilize simplified lignin model compounds and investigate the very limited number of ligninolytic genes in metagenomes, highlighting the currently limited knowledge of ligninolytic enzymes (2, 3, 36). Moreover, the functional and ecological traits of bacterial decomposer populations for different lignin/lignocellulose substrates have not been

thoroughly explored, although there is some evidence for a differential preference by bacteria for fresh versus aged TerrOC (37). In addition, bacterial specialization in the degradation of lignin, cellulose, and hemicellulose is known to occur (2). Interestingly, the availability of labile organic matter (e.g., cellulose and hemicellulose) was observed to promote the degradation of recalcitrant organic matters (e.g., lignin) and is known as the priming effect (PE) (38, 39). This suggested that complementation among lignin, cellulose, and hemicellulose decomposers might contribute to lignocellulose degradation. However, the mechanism of PE has not been well-characterized, especially in coastal wetlands. Coastal wetlands feature strong dynamic alternations between aerobic and anaerobic micro-environments. It's known that bacteria employ different strategies to catabolize lignin and cellulose under aerobic and anaerobic conditions (40–42). Whether, and how, interactions between aerobic and anaerobic functional genes contribute to PE is largely unknown.

Here, we utilized an *in situ* enrichment experiment to investigate the bacterial community compositions, mechanisms of community assembly and functional genes for different native lignocellulosic substrates and lignin over an 18 months exposure period in the south-eastern intertidal zone of Zhairuoshan Island, Zhejiang, China. We aimed to address current knowledge gaps to advance our understanding of coastal bacterial communities driving lignocellulose degradation. (i) How do substrates and exposure times govern bacterial community assembly and composition? (ii) What is the genomic machinery to degrade lignocellulose? (iii) Do the taxonomic groups and functional genes involved display a substrate-dependent pattern? (iv) What is the mechanism of PE in coastal intertidal wetlands? Our study provides new insights into coastal bacterial consortia with the great metabolic potential for various lignocellulose substrates, challenging the current view of lignin stability within marine ecosystems.

## RESULTS

### Lignocellulose degradation by enriched coastal bacterial communities

We performed an *in situ* enrichment experiment in the coastal intertidal wetlands of the East China Sea to enrich bacterial lignocellulose degrading communities (Fig. 1A) (43). Four different substrates, aspen lignocellulose (hardwood), pine lignocellulose (softwood), rice straw (herbaceous), and Norway spruce lignin (Sigma-Aldrich, catalog# 370959), were sampled at four different time points (3, 6, 12, and 18 months) over an 18 months period, based on the substrate degradation rates (Fig. 1; see also Table S1 and S2 at https://doi.org/10.5281/zenodo.7659223). In total 96 bacterial consortia, 16 groups with six replicates each, were obtained. Each group exhibited the capacity of substrate degradation, which increased with longer exposure times, reaching a peak at 18 months (Fig. 1C through F; see also Table S2 at https://doi.org/10.5281/zenodo.7659223). The consortia of aspen and pine substrates exhibited similar degradation values (e.g., ~35% lignin consumption at 18 months), while rice straw consortia showed a maximum (e.g., ~63% lignin consumption at 18 months) during the enrichment period. This discrepancy was likely related to the substrate properties. Compared to rice straw, the aspen and pine substrates have similar compositions, with higher lignin contents (~25%, Fig. 1B). It is worth noting that more lignin from lignocellulose substrates (aspen, pine, and rice straw) was consumed than that of purified Norway spruce lignin, demonstrating that synergistic effects should occur among lignin, cellulose, and hemicellulose degradation (see Table S2 at https://doi.org/10.5281/zenodo.7659223).

### Effects of the enrichment periods and substrates on bacterial taxonomic community assembly

Consistent with the increased DNA extraction yields over time, the alpha diversity of the communities, including richness and evenness, mostly increased over time, the exception being the consortia enriched on rice straw at 12 months (Fig. 2A; also see Fig. S1 at https://doi.org/10.5281/zenodo.7659223). On one hand, richness (the number of unique

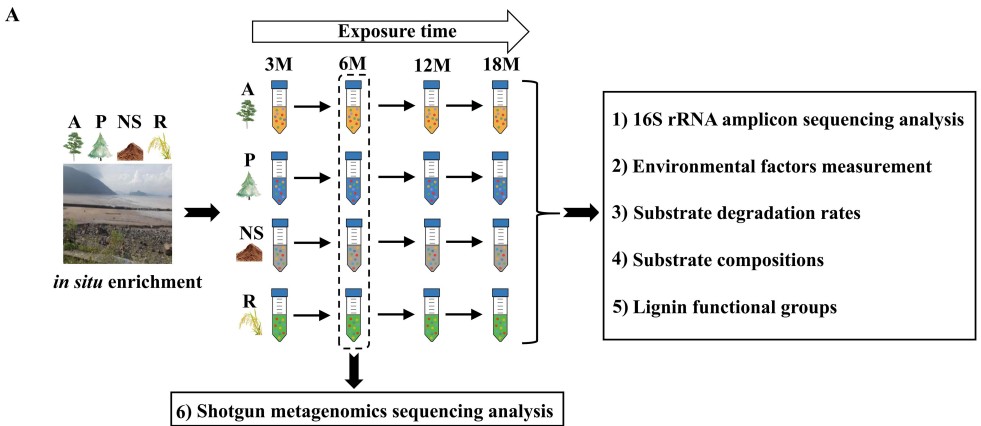

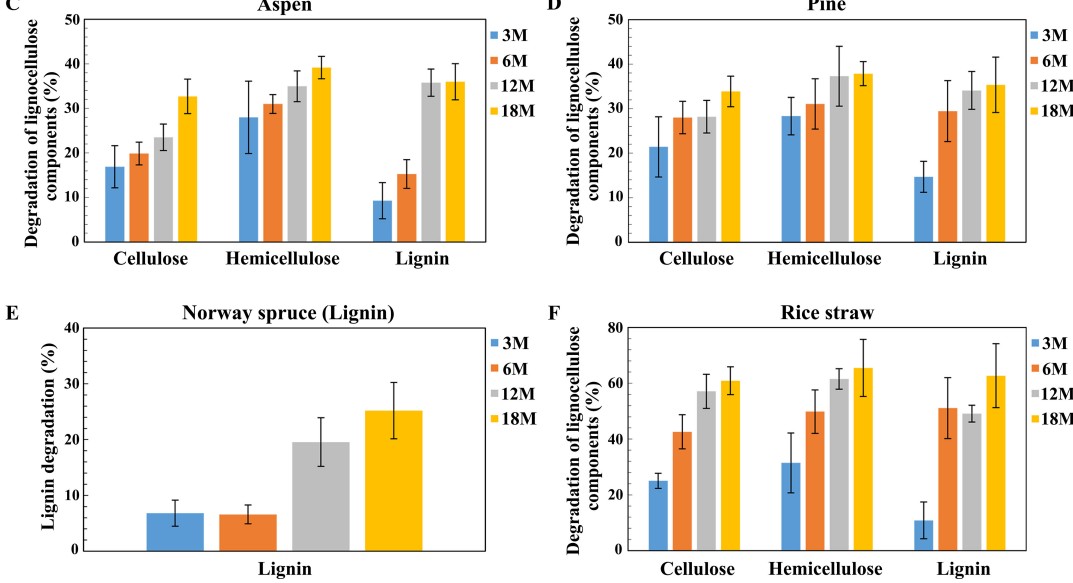

**FIG 1** *In situ* lignocellulose enrichment experiment. (A) Schematic representation of the experiment design. A: aspen, *P*: pine, NS: Norway spruce lignin, R: rice straw. (B) Lignocellulosic composition of each substrate. (C–F) Percentage of consumed lignin, cellulose, and hemicellulose in each substrate by the enriched bacteria consortia over time. 3 M–18 M: different amounts of exposure time. M, months. Significant differences between samples are indicated by asterisks (*, $P < 0.05$, **, $P < 0.01$).

amplicon sequence variants [ASV]) significantly increased ~3.5-fold from 3 to 18 months (Wilcoxon rank-sum test, see Fig. S1C at https://doi.org/10.5281/zenodo.7659223). On the other hand, a more even distribution pattern was observed over time (see Fig. S1D at https://doi.org/10.5281/zenodo.7659223). *Sulfurovum*, as the most dominant genus (19.11%–32.32%) at 3 months, was reduced to 6.16%–15.60% at 18 months. In contrast, *Pseudodesulfovibrio* (0.52%–4.27%), *Marispirochaeta* (0.09%–0.30%), and *Desulfosarcina* (0.01%–0.14%), as minor genera at 3 months, increased to 3.74%–8.17%, 1.74%–2.82%, and 3.18%–3.34%, respectively, at 18 months (see Fig. S2A and B at https://doi.org/10.5281/zenodo.7659223). This indicated that the consortium members were amplified and more evenly distributed over time (see Fig. S1C and D at https://doi.org/10.5281/zenodo.7659223). Interestingly, with the increased consortium diversity over time,

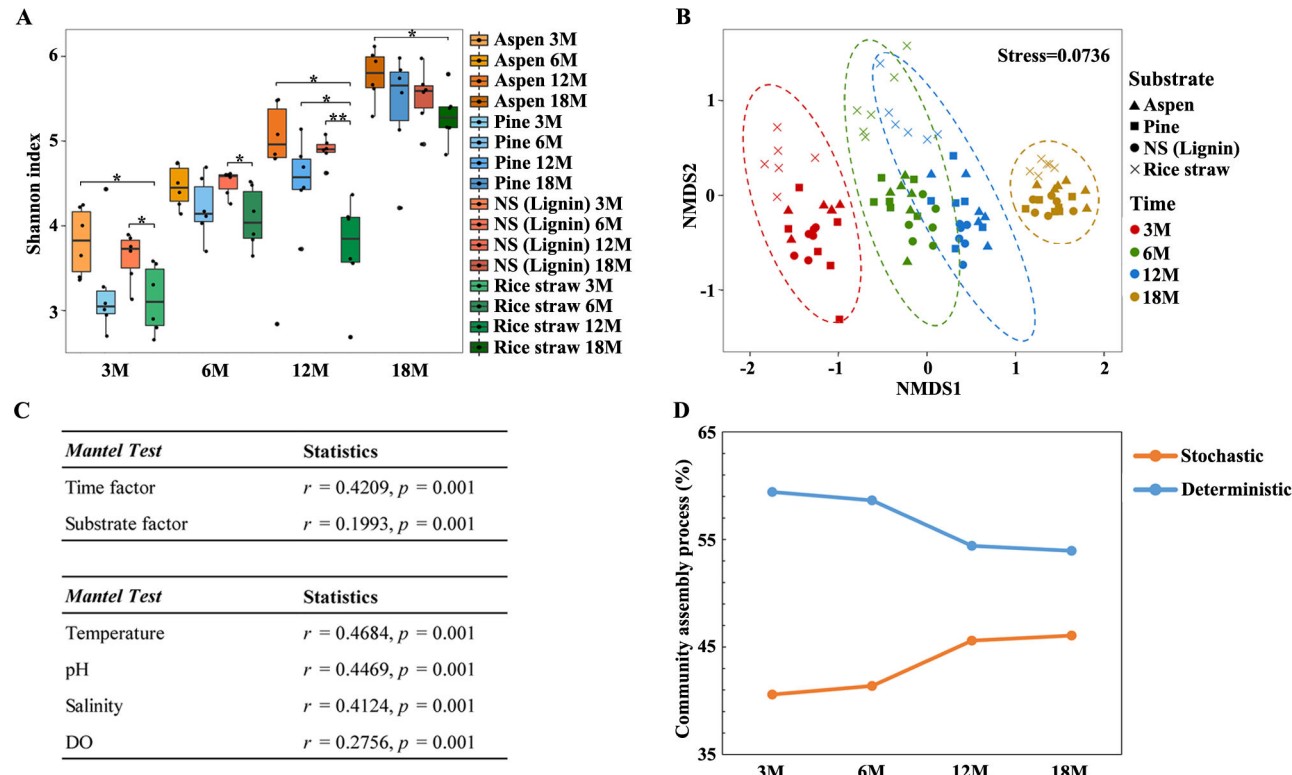

**FIG 2** The diversity and assembly mechanism of the enriched bacterial communities. (A) Shannon diversity of the bacterial community over enrichment time. Significant differences between samples are indicated by asterisks (*$P < 0.05$, **$P < 0.01$). (B) An NMDS profile of the 96 bacterial taxonomic communities based on the Bray–Curtis dissimilarities. Ellipses indicate the 95% CI grouping effects of exposure time, as represented by different colors. Stress is included in the upper right corner. (C) The association of environmental factors and bacterial communities. The compositional variations for taxonomic groups were analyzed. Mantel's r was calculated for association strength. (D) Dynamic changes of stochasticity and determinism during the succession of bacterial communities.

correlation among the ASVs were enhanced, indicating interactions among taxa were enhanced with the increasing length of the enrichment period (see Fig. S2C and D at https://doi.org/10.5281/zenodo.7659223). Nonmetric multidimensional scaling (NMDS) analysis showed a temporal separation of consortia across all substrates (Fig. 2B). Coinciding with the four time points, the consortia were grouped into four clusters. Moreover, a clear time dependent dissimilarity pattern was observed for consortia across all substrates, where community dissimilarities were enhanced over time (see Fig. S3 at https://doi.org/10.5281/zenodo.7659223). To discern the respective contributions of different taxa to the meta-community temporal pattern, time-dissimilarity relationships of the top 20 most abundant taxa were examined via Spearman's rank correlation analysis (see Fig. S3 at https://doi.org/10.5281/zenodo.7659223). *Labilibaculum*, *Pseudodesulfovibrio*, *Sulfurovum,* and *Desulfosarcina* displayed a significant correlation between community dissimilarity and temporal distance. This indicated they were likely sensitive to exposure times, possibly due to habitat preference (e.g., temperature and pH) (see Fig S3F through I at https://doi.org/10.5281/zenodo.7659223). In contrast, the correlations of community dissimilarity-temporal distance were decreased in *Vibrio*, *Arcobacter*, *Cohaesibacter,* and *Bacteroides* (see Fig. S3J through M at https://doi.org/10.5281/zenodo.7659223), suggesting that they can resist environmental perturbation. Overall, exposure times exerted significant pressure on community assembly.

In addition, substrates also impacted community assembly, with substrate-specific differences in alpha diversity observed. The consortia enriched by woody substrates (aspen, pine, and Norway spruce lignin) showed significantly higher Shannon index, while communities enriched by herbaceous substrate (rice straw) presented the lowest Shannon index at 12 and 18 months (Fig. 2A). Moreover, beta diversity showed clear

substrate-specific differences for consortia across substrates, with the exception of aspen and pine. Mantel test further suggested substrate composition (cellulose/hemicellulose/lignin) showed significant correlation with the community variance (r = 0.25, *P* = 0.001). Such dissimilarity presented dynamic fluctuation with community succession. Communities at 3 months initially exhibited substrate dependent variance and peaked at 6 months. Subsequently, communities across substrates gradually converged from 12 to 18 months (see Fig. S3N at https://doi.org/10.5281/zenodo.7659223). Together, this confirmed that substrates played important roles in community assembly, but the effect was reduced over time, possibly due to substrate consumption.

The results suggested that exposure time and substrates governed community assembly. To clarify how well community composition and variance could be affected by substrates and exposure time, Mantel test were performed. Variance in the composition was primarily correlated with the enrichment period, followed with the substrates (Fig. 2C). The *in situ* enrichment period was the comprehensive factor, which not only involved the exposure time, but also related the environmental factors. Hence, Mantel test was also used to assess the effects of environmental factors, including pH, temperature, salinity, and DO. All of these factors significantly associated with the compositional variations of bacterial consortia (*P* = 0.001, Fig. 2C). Among them, temperature was the most influential factor on taxonomic composition, followed by pH, salinity, and DO, respectively (Fig. 2C). These results suggested that deterministic processes should govern community assembly. To verify this, null model analysis was performed to investigate the relative importance of deterministic and stochastic processes in the assembly of consortia. The analysis showed that homogenous selection (a deterministic process) was the most important process, accounting for 53.9%–59.2% of the community variation across all communities, followed by stochastic processes, such as drift (18.1%–30.8%) and dispersal limitation (15.0%–22.1%, see Fig. S4A at https://doi.org/10.5281/zenodo.7659223). This confirmed that deterministic processes contributed a higher proportion of community variation than stochastic processes (Fig. 2D). Furthermore, the contributions of deterministic processes varied with the substrate, exhibiting different selection pressures among the different substrates. A higher proportion of homogenous selection was observed in consortia enriched on Norway spruce lignin and aspen (71.9%–67.9% at 3 months), compared to consortia enriched on rice and pine (45.4%–54.6% at 3 months, see Fig. S4 at https://doi.org/10.5281/zenodo.7659223). In addition, the impact of stochastic processes was enhanced over time with the consumption of substrate (Fig. 2D).

## Distinct compositions of taxonomic groups and functional traits among various lignocellulose enrichments

As stated earlier, differences of the communities enriched by different carbon sources were observed at each time point and reached the peak at 6 months (see Fig. S3N at https://doi.org/10.5281/zenodo.7659223). To further clarify substrate-dependent populations, a cladogram of the LEfSe analysis was constructed (Fig. 3). Rice straw at 6 months recruited more of the genera *Sulfurovum* (45.22%), *Labilibaculum* (8.52%), *Cohaesibacter* (7.02%), *Marispirochaeta* (5.90%), *Bacteroides* (5.22%), *Maribellus* (2.98%), and *Petrimonas* (0.05%). In contrast, Norway spruce lignin at 6 months significantly enriched *Vibrio* (14.17%), *Arcobacter* (6.91%), *Thiolapillus* (4.22%), *Anaerofustis* (2.75%), *Desulfovibrio* (2.47%), *Shewanella* (2.35%), *Fulvivirga* (1.87%), *Desulfopila* (1.48%), *Alkalispirochaeta* (0.10%), *Winogradskyella* (0.01%–0.05%), *Emcibacter* (0.05%), *Enterococcus* (0.05%), and *Reichenbachiella* (0.04%), while aspen/pine substrates highly enriched *Pseudodesulfovibrio* (8.43%), *Spirochaeta* (6.49%), *Marinilabilia* (5.44%), and *Mangrovibacterium* (2.26%) (see Table S3 at https://doi.org/10.5281/zenodo.7659223).

Next, the metagenomes of the consortia with the largest taxonomic differences, enriched at 6 months, were sequenced to reflect the compositions of functional traits in response to different substrates (Fig. 1A; see also Fig. S3N at https://doi.org/10.5281/zenodo.7659223). A total of 548 gene families were identified to participate in

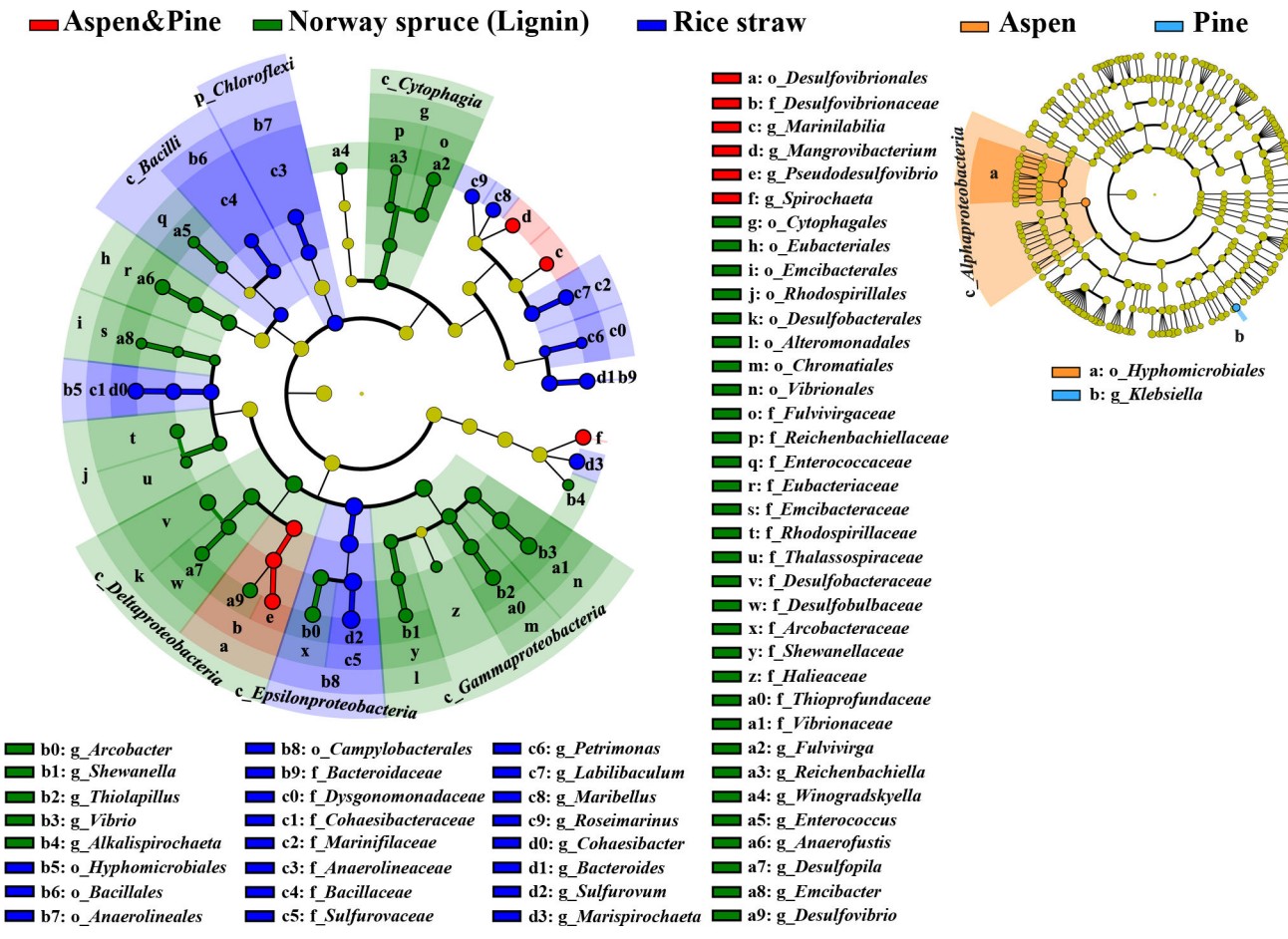

**FIG 3** The substrate-specific populations under different substrates. Significantly discriminant taxa among different substrates, from phylum to genus level. Kruskal–Wallis test, $P < 0.05$, and LDA score = 3.5.

lignocellulose degradation and could be divided into cellulose hydrolysis (67 gene families), hemicellulose (87 gene families), and lignin degradation (394 gene families, see Table S4 at https://doi.org/10.5281/zenodo.7659223). Interestingly, the compositions of gene families encoding cellulases were similar among the four substrates ($P > 0.05$), as indicated by three different complementary nonparametric tests: permutational multivariate analysis of variance (Adonis), analysis of similarity (ANOSIM) and multiple response permutation procedure (MRPP) (see Fig. S5 at https://doi.org/10.5281/zenodo.7659223). In contrast, gene families involved in hemicellulose and lignin degradation showed significant differences ($P < 0.05$) between consortia enriched on woody and herbaceous substrates (see Fig. S5 at https://doi.org/10.5281/zenodo.7659223). The difference was especially significant for lignin degrading genes ($P < 0.01$, see Fig. S5C and D at https://doi.org/10.5281/zenodo.7659223), suggesting the specialization for woody and herbaceous lignin degradation.

Thus, functional genes involved in lignin degradation were further examined. The relative abundance of *dypB*, the most abundant ligninolytic gene in all samples, was significantly higher in consortia under woody lignin, compared to consortia under herbaceous lignin ($P < 0.05$). In contrast, *dypA*, as the second abundant ligninolytic gene, showed higher abundance in consortia enriched on rice straw (herbaceous lignin) ($P < 0.05$, Fig. 4; see also Fig. S6A at https://doi.org/10.5281/zenodo.7659223). Additionally, genes encoding laccase (44) and manganese superoxide dismutase (MnSOD, a novel bacterial ligninolytic enzyme [45]) were also detected in all samples, although their abundances were lower than *dyps* (see Fig. S6A at https://doi.org/10.5281/zenodo.7659223). Hence, lignin depolymerization may be mainly mediated by DyPs in

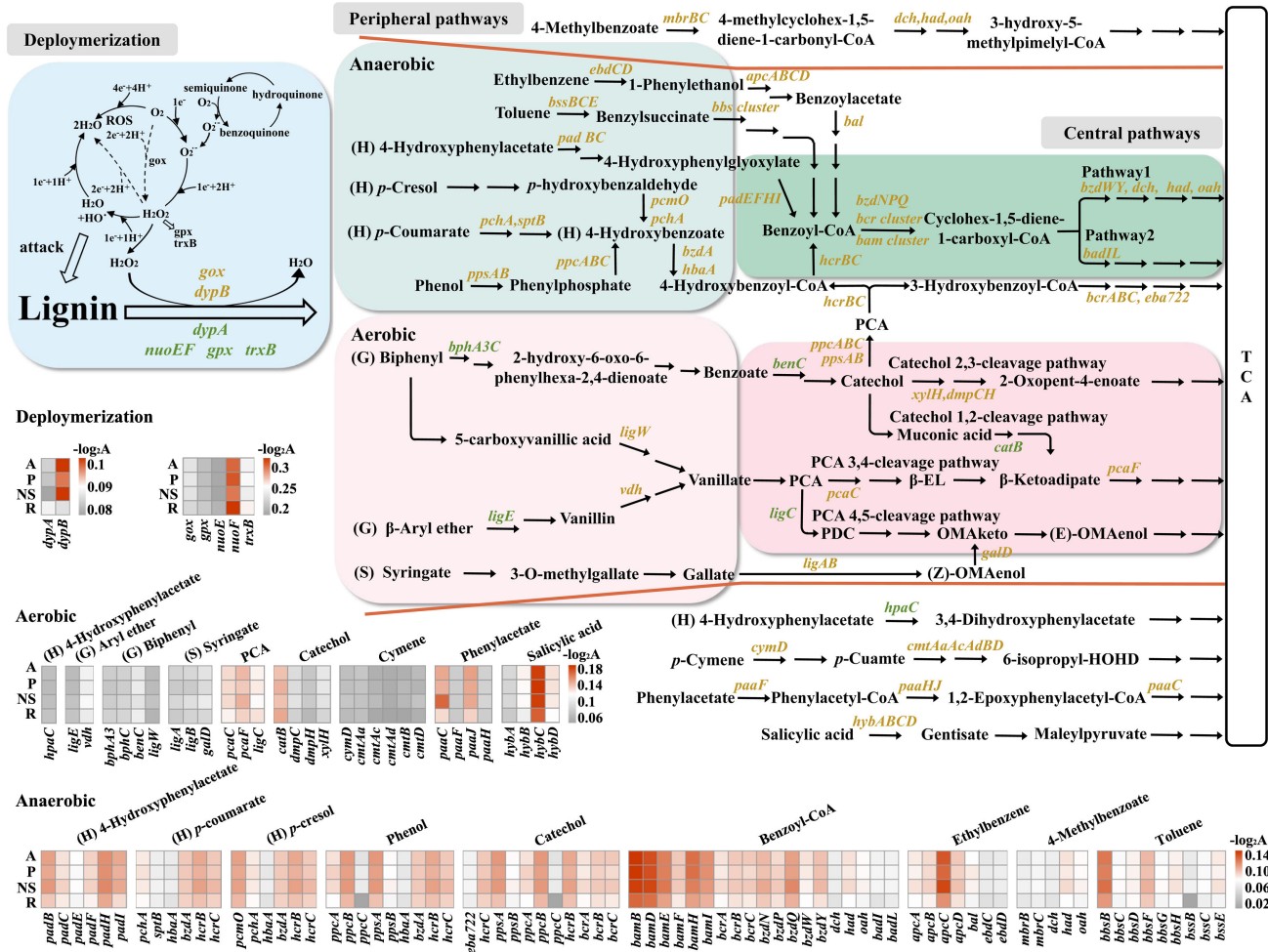

**FIG 4** The abundance differences of genes involved in aerobic/anaerobic lignin degradation pathways between consortia under woody and herbaceous lignin substrates. Orange genes showed significantly higher abundance in consortia enriched on woody substrates, while green genes exhibited significantly higher abundance in consortia enriched on the herbaceous substrate. The relative abundances of genes are visualized as heatmaps. −Log$_2$A represents base 2 logarithm-transformed gene reads levels as indicated by normalized reads counts. NS, Norway spruce lignin.

these consortia, supplemented with laccase and MnSOD (see Fig. S6A at https://doi.org/ 10.5281/zenodo.7659223). Nineteen gene families encoding auxiliary enzymes were also identified in these samples, which were reported to assist ligninolytic enzymes in lignin oxidation (see Table S4 at https://doi.org/10.5281/zenodo.7659223) (46–48). Their relative abundances also exhibited substrate-dependent pattern. Genes encoding quinone reductase (*nuoE* and *nuoF*) and enzymes in the reactive oxygen species (ROS), e.g., thioredoxin reductase (*trxB*) and glutathione peroxidase (*gpx*), showed higher abundance in consortia enriched on rice straw, while *gox*, encoding glycolate oxidase, were highly enriched under woody lignin (*P* < 0.05, Fig. 4; also see Fig. S6B and C at https://doi.org/10.5281/zenodo.7659223). Taken together, consortia enriched on herbaceous lignin might preferentially employ DypA, QR, and ROS enzymes for lignin oxidation. In contrast, the consortia enriched on woody lignin were likely to mobilize DypB and glycolate oxidase for lignin depolymerization (Fig. 4).

Next, comprehensive functional traits involved in degradation pathways of lignin-derived aromatic compounds were identified in all consortia, including 46 anaerobic/aerobic peripheral pathways and seven anaerobic/aerobic ring cleavage pathways (Fig. 4; also see Fig. S7 at https://doi.org/10.5281/zenodo.7659223). This reflected the great metabolic potential of coastal consortia to degrade various lignocellulose substrates. Interestingly, genes in anaerobic pathways showed higher abundance

than those of aerobic pathways, with the exception of the 4-hydroxypenylacetate, phenylactate, and phthalate degradation pathways, suggesting anaerobic catabolism should play a key role in lignin degradation in coastal intertidal wetlands. Consistent with the contents of guaiacyl (G), syringyl (S), and *p*-hydroxyphenyl (H) types lignin units in all substrates (see Table S1 at https://doi.org/10.5281/zenodo.7659223), genes involved in H-type lignin units showed the greatest abundance, followed by genes taking part in G-type lignin degradation and S-type lignin degradation, respectively (see Fig. S7A at https://doi.org/10.5281/zenodo.7659223). Importantly, substrate-specific preferences were also observed. The consortia enriched on herbaceous lignin had a higher abundance of genes involved in aerobic G-type (biphenyl) and H-type (4-hydroxypenylacetate) lignin catabolism (Fig. 4; see also Fig. S7B at https://doi.org/10.5281/zenodo.7659223). In contrast, the consortia enriched on woody lignin substrates showed higher abundance of genes involved in anaerobic H-type lignin catabolism, e.g., 4-hydroxypenylacetate and *p*-coumarate degradation pathways, and aerobic S-type lignin degradation (syringate) (Fig. 4; see also Fig. S7B and C at https://doi.org/10.5281/zenodo.7659223). Moreover, the consortia utilizing woody lignin substrates commonly exhibited higher gene abundances in other lignin-derived aromatic compound degradation pathways, including aerobic *p*-cymene, phenylacetate, and salicylic acid, as well as anaerobic 4-methylbenzoate and phenol degradation pathways (Fig. 4; also see Fig. S7 at https://doi.org/10.5281/zenodo.7659223).

Following degradation in the peripheral pathways, these aromatic compounds are converted into protocatechuate (PCA) and catechol under aerobic conditions and benzoyl-CoA under anaerobic conditions. Subsequently, these compounds are processed through the central ring cleavage pathways and finally incorporated into the tricarboxylic acid cycle to provide energy for the consortia (49). Genes involved in central ring cleavage pathways were identified in all consortia, including three PCA degradation pathways (4,5-, 2,3-, and 3,4-cleavage pathways), two catechol degradation pathways (1,2- and 2,3-cleavage pathways) and two benzoyl-CoA degradation pathways (Fig. 4; see also Table S4 at https://doi.org/10.5281/zenodo.7659223). Furthermore, genes involved in PCA 3,4-cleavage pathway, catechol 1,2-cleavage pathway, and benzoyl-CoA pathway 1 (*bam* gene cluster) exhibited the greater abundance, compared to other homogenous branched pathways. This suggested a preference in the coastal wetland community. Interestingly, the relative abundance of genes involved in two anaerobic benzoyl-CoA pathways and the aerobic catechol 2,3-cleavage pathway were higher in consortia from woody lignin, compared to consortia from herbaceous lignin (Fig. 4). This further indicated the specialization between woody lignin and herbaceous lignin degradation.

## Synergistic effects on lignocellulose degradation

As stated earlier, lignin consumption was significantly higher in the lignocellulosic substrates than that in the purified lignin substrate (Fig. 1; see also Table S2 at https://doi.org/10.5281/zenodo.7659223). This suggested that there was a synergistic effect between lignin and hemi-/cellulose degradation. To reveal the synergistic relationship of lignocellulose degraders, taxon co-occurrence network analysis was performed, combined with correlation analysis between substrate composition and community (see Fig. S8A and B at https://doi.org/10.5281/zenodo.7659223). As a result, a network was constructed with 288 nodes and 5,699 edges (Spearman's $|r| > 0.6$ and *P* value $< 0.05$, see Fig. S8B at https://doi.org/10.5281/zenodo.7659223). On one hand, lignin degradation was positively correlated with consortium diversity, as the network showed that the 60 most abundant ASVs in the network were involved in lignin degradation, and were mainly represented by *Vibrio* and *Arcobacter*. In contrast, 43 ASVs were related to cellulose degradation, e.g., *Pseudodesulfovibrio* an*d Oceanispirochaeta*, while 27 ASVs were correlated with hemicellulose degradation, e.g., *Sulfurovum* and *Labilibaculum* (see Fig. S8A and B at https://doi.org/10.5281/zenodo.7659223). Meanwhile, high interconnection among nodes (ASVs), involved in cellulose, hemicellulose, and lignin degradation, was observed (see Fig. S8B at https://doi.org/10.5281/zenodo.7659223).

One hundred sixteen edges connected 23 ASVs involved in cellulose degradation with 24 lignin degradation ASVs, whereas 58 edges connected 11 hemicellulose degradation ASVs and 20 ASVs in lignin degradation. In addition, 59 edges connected 27 ASVs related to cellulose degradation and 18 ASVs correlated with hemicellulose degradation. Together, this suggested that cooperation among these ASVs, which should be responsible for lignin, cellulose, and hemicellulose decomposition, occurred for efficient lignocellulose degradation.

Next, to further clarify the synergistic mechanism, a co-occurrence network of functional genes was constructed (see Fig. S8C at https://doi.org/10.5281/zenodo.7659223). First, genes involved in lignin depolymerization and aromatic compound catabolism were closely connected. Ligninolytic oxidative genes displayed positive and negative correlation with genes in lignin-derived aromatic compound degradation pathways (see Fig. S9A at https://doi.org/10.5281/zenodo.7659223). (i) The laccase gene (*lac*) was positively correlated with *catE* in the aerobic catechol meta-cleavage pathway. (ii) *dypA* was positively connected to *mhpT* (in aerobic 3-hydroxycinnamic acid pathway), while negatively linked to *cmtAA* (in aerobic cymene meta-cleavage pathway). (iii) *dypB* was positively linked to *mhbH* (in aerobic salicylic acid degradation pathway). Interestingly, no links among aerobic ligninolytic oxidative genes were observed, indicating they were likely to independently depolymerize lignin, presenting an aggregation mode for lignin depolymerization. Moreover, these ligninolytic oxidative genes were negatively correlated with genes in anaerobic aromatic compound pathways. For instance, the *dypA* and *lac* genes were negatively connected to genes in the anaerobic catechol, benzoyl-CoA, and 4-hydroxybenzoate degradation pathways. This demonstrated specialization between aerobic lignin depolymerization and anaerobic lignin-derived aromatic compound degradation in the coastal wetland, which possibly presents a competition interaction.

Second, aerobic ligninolytic, cellulase, and hemicellulase genes were highly connected with each other (see Fig. S9B at https://doi.org/10.5281/zenodo.7659223). Two hundred three links were observed between 38 cellulose hydrolysis genes and 67 hemicellulose hydrolysis genes. These highly interconnected cellulases and hemicellulases were mostly clustered in the GH5, GH3, GH16, and GH43 families. The GH3 family is involved in cellulose hydrolysis, while the GH16 (with versatile endoglycosidase activity) and GH43 (as xylan degraders) families are involved in hemicellulose hydrolysis (50, 51). In addition, the GH5 family participates in both cellulose and hemicellulose hydrolysis (52). Meanwhile, 14 ligninolytic genes were connected to 13 cellulase genes and 23 hemicellulase genes, respectively. *dypA* was positively connected with genes encoding GH5_22, GH43_27, GH43_22, and GH16_21, while *dypB* was correlated with genes encoding GH16_16. In addition, laccase genes were connected to GH43_6 genes, whereas MnSOD genes were linked to GH5_26 genes. Hence, the functional genes, which specialize in aerobic lignin, cellulose, and hemicellulose decomposition, worked together to synergically degrade lignocellulose.

Third, genes involved in anaerobic lignocellulose degradation were closely interconnected, but presented different modes with aerobic lignocellulose degradation (see Fig. S10 at https://doi.org/10.5281/zenodo.7659223). Genes encoding cellulosomes (the multienzyme complexes for anaerobic hemi-/cellulose hydrolysis [53]) were positively interconnected. These contained cellulose-binding protein families (CBM2 and CBM3), cellulases (GH5, GH9, and GH48), and hemicellulases (GH10, GH11, and GH43). The genes involved in anaerobic aromatic compound metabolism were also highly positively interconnected (see Fig. S10 at https://doi.org/10.5281/zenodo.7659223). Interestingly, these two groups exhibited mostly negative connections (152 negative links versus 35 positive links). This not only indicated specialization between anaerobic hemi-/cellulose hydrolysis and anaerobic aromatic compound degradation, but also revealed competitive interactions (Fig. 5; see also Fig. S10A at https://doi.org/10.5281/zenodo.7659223).

Finally, integrating all of the above-stated information, a synergistic model for lignocellulosic degradation was proposed, which identified the potential taxonomic

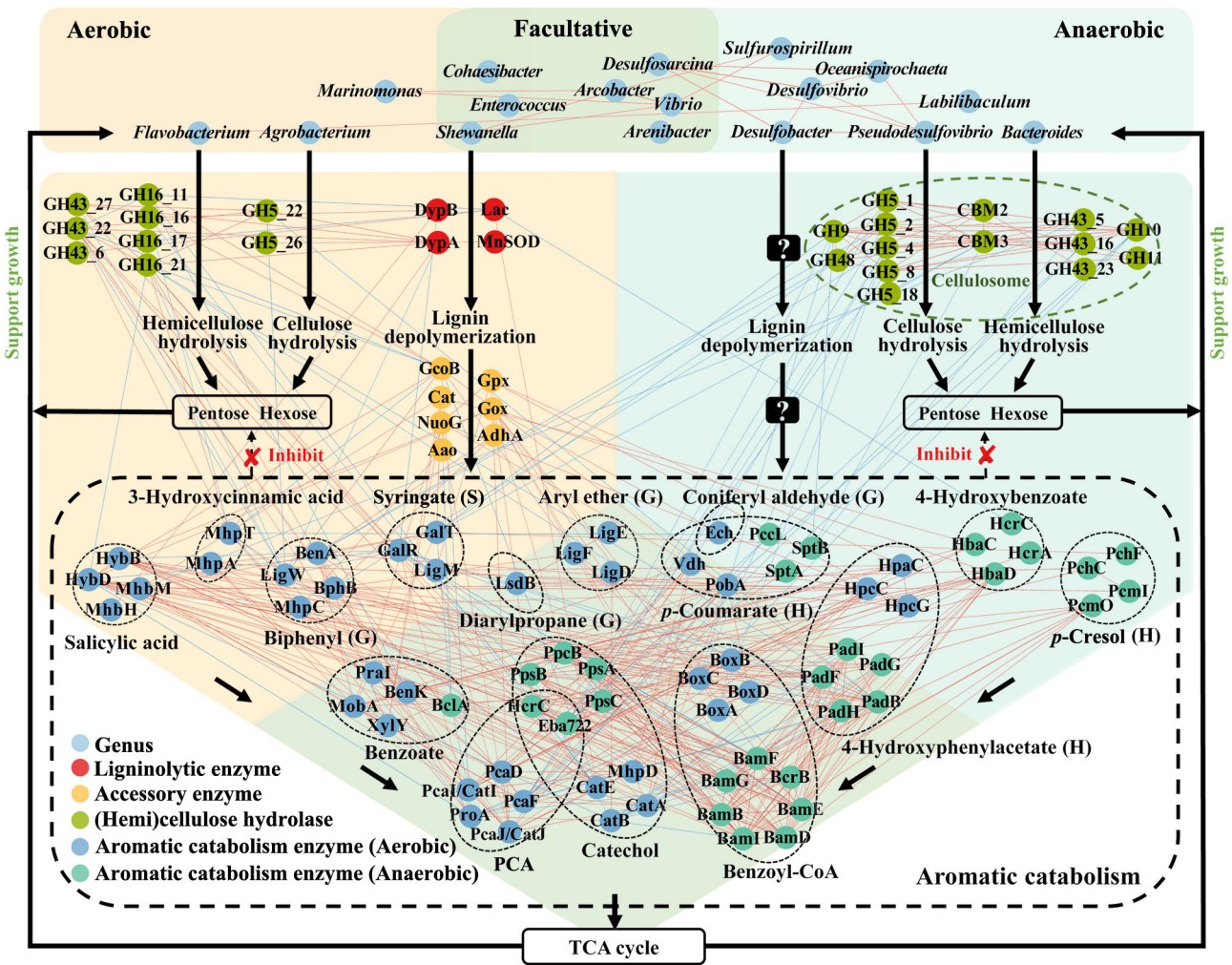

**FIG 5** Priming effect mechanism of bacterial consortia in the coastal intertidal zone. The potential biological actors at the levels of taxa and functional genes between aerobic and anaerobic conditions are indicated. The hemi-/cellulose degraders release carbohydrate monomers supporting consortia growth. Enhanced cell abundances contribute to the secretion of ligninolytic enzymes to promote lignin depolymerization and further accelerate the hemi-/cellulose exposure for the hemi-/cellulose degraders. Moreover, the aromatic compound degraders degrade aromatic compounds, which not only provide energy and carbon sources for themselves, but also alleviate the toxicity for the consortia. Question marks represent unknown functional genes involved in anaerobic lignin depolymerization. Red link: positive connection, blue link: negative connection.

groups and functional genes (Fig. 5). Anaerobic (e.g., *Sulfurospirillum*, *Bacteroides*, *Desulfovibrio*, *Pseudodesulfovibrio,* and *Labilibaculum*) and facultative bacteria (e.g., *Desulfosarcina*, *Arcobacter* and *Arenibacter*) encode cellulosomes for hemi-/cellulose hydrolysis, while aerobic (e.g., *Flavobacterium* and *Agrobacterium*) and facultative members (e.g., *Vibrio* and *Shewanella*) encode free cellulases for hemi-/cellulose degradation. The interconnection at taxon and gene levels suggested that these potential degraders could work together to efficiently hydrolyze hemi-/cellulose via the strategies of aggregation and division of labor (e.g., aerobic and anaerobic hydrolysis). The released carbohydrate monomers could support growth of the consortia, including themselves and members associated with lignin degradation (e.g, aerobic *Marinomonas*, anaerobic *Desulfobacter* and facultative *Cohaesibacter*, *Vibrio*, *Arcobacter*, *Desulfosarcina*, *Shewanella,* and *Enterococcus*). Hence, higher 16S rRNA copy numbers were observed under lignocellulose substrates, compared to that under the pure lignin substrate (see Fig. S1B at https://doi.org/10.5281/zenodo.7659223). Enhanced cell abundance would contribute to the secretion of ligninolytic oxidative enzymes and accessory enzymes (e.g., DypB and MnSOD in *Vbrio* and *Shewanella*, and DypA in *Cohaesibacter*), which promote lignin

depolymerization to release various aromatic compounds and further accelerate hemi-/cellulose exposure. This was supported by the close connections between genes encoding lignin depolymerase and hemi-/cellulases (Fig. 5). Subsequently, the various aromatic compounds can be further catabolized by consortia (Fig. 5). For instance, *Marinomonas* and *Shewanella*, both of which encode the *lsdB*, *ligE-D,* and *ech* gene clusters, could aerobically degrade G-type lignin units (diarylpropane, aryl_ether, and coniferyl aldehyde), respectively, while *Cohaesibacter*, *Vibrio,* and *Arcobacter* all encode the *bph*, *gal,* and *hca* gene clusters for aerobic metabolism of G-type lignin unit (biphenyl), S-type lignin unit (syringate), and 3-hydroxycinnamic acid, respectively. Meanwhile, *Agrobacterium* and *Flavobacterium* encode the *hpc* and *vdh* gene clusters involved in aerobic H-type lignin unit degradation pathways, e.g., 4-hydroxyphenylacetate and *p*-coumarate, whereas *Vibrio* and *Arcobacter* encode the *pad*, *spt,* and *pcm* gene clusters for anaerobic degradation of H-type lignin units, e.g., 4-hydroxyphenylacetate, *p*-coumarate, and *p*-cresol. Such aggregation and division of labor could enhance aromatic compound metabolism. Importantly, close connections between aerobic and anaerobic pathways of the same metabolites were observed (Fig. 5). This suggests that aerobic metabolites, e.g., catechol and PCA, could be further metabolized under anaerobic conditions and, in turn, anaerobic metabolites, e.g., benzoyl-CoA, can also be degraded under aerobic conditions. Such cross-feeding cooperation (54, 55) would stimulate aromatic compound metabolism under fluctuating micro-environments. These compounds are then funneled into the TCA cycle to provide energy and alleviate the inhibition of hemi-/cellulose hydrolysis.

## DISCUSSION

### The different impacts of substrates and exposure times on long-term community assembly

The substrate-dependent enriched bacterial communities and substantial lignocellulose degradation supported the notion that bacterial consortia contribute to lignocellulose degradation in coastal intertidal wetlands. Moreover, our results showed that deterministic processes played important roles in determining the community assembly. Interestingly, the homogenous selection had greater control of community composition under aspen and Norway spruce lignin than that of pine and rice during the early stage, suggesting there were different selection pressures among these substrates (see Fig. S4 at https://doi.org/10.5281/zenodo.7659223). Combined with previous research, in which stochasticity drove early coastal substrate enriched community assembly (7–14 days) and deterministic factors became increasingly important for late community assembly (19–56 days) (31), a conceptual model of substrate driven bacterial consortium assembly process was proposed. Initial communities, within several days, are commonly driven by stochastic processes with the inherent randomness of dispersal and ecological drift (31, 32). Next, selection pressures (e.g., substrates, pH and salinity) shaped communities over the next 3–6 months, by selecting for specific organisms. Following settlement, consumed substrates trigger stochastic processes to rise again (12–18 months), while deterministic processes still retain a major role (Fig. 2D).

On one hand, the varied organic carbon substrates acted as environmental filters for each community's composition. Pine/aspen wood, Norway spruce lignin, and rice straw substrates, with their distinctly varied compositions, enriched significantly different bacterial communities. Notably, consortia enriched on woody lignocellulose showed higher diversity than consortia on herbaceous substrate, possibly due to the higher content of recalcitrant lignin in woody lignocellulose. It has been reported that higher functional diversity increased lignin degradation (56), consistent with that most ASVs in the network as potential lignin degraders (see Fig. S8 at https://doi.org/10.5281/zenodo.7659223). Consortia with greater metabolic diversity and microbial interactions should contribute to synergistic lignin catabolism without metabolite limitation or repression, due to the inherent heterogeneity of lignin.

On the other hand, exposure time, a comprehensive factor, had a greater effect on community assembly. Environmental factors (e.g., temperature, pH, salinity, and DO), which are embedded within the enrichment period, significantly affected community assembly (Fig. 2C). This was consistent with previous reports (28, 57, 58), confirming the importance of environmental variables in community variation for coastal intertidal wetlands. Surprisingly, a clear community separation according to temporal variation was revealed in this study (Fig. 2B). This was inconsistent with a previous report which suggested that benthic archaeal community compositions did not display significant seasonal differences (59). Such a disparity demonstrates a greater temporal habitat fluctuation in the intertidal zone than in sediments. Meanwhile, this was different from previous observations in microbial communities from coastal marine systems, which showed periodic patterns with ~6-month oscillations (60, 61). In our study, the temporal separation presented an amplified pattern, with increased alpha diversity over time. This was more likely due to an inherent property of lignocellulose substrates, such as an increased roughness and accumulated degradable organic carbon (e.g., mono-/oligo-saccharides and aromatic compounds) with substrate depolymerization (31). Furthermore, a time dependent dissimilarity pattern was observed, revealing a higher rate of bacterial community turnover in the coastal intertidal zone (see Fig. S3 at https://doi.org/10.5281/zenodo.7659223). The contribution of different bacterial taxa to the whole-community time-dissimilarity pattern was distinct, as exemplified by *Sulfurovum*, *Vibrio,* and *Arocbacter* (see Fig. S3 at https://doi.org/10.5281/zenodo.7659223). *Sulfurovum*, with the greatest abundance, showed the highest cross-sample community similarities across all samples (see Fig. S2A and B at https://doi.org/10.5281/zenodo.7659223). However, it also exhibited a stronger time-dissimilarity relationship. In contrast to *Sulfurovum*, *Vibrio,* and *Arocbacter*, which showed lower abundances, and the second and third highest cross-sample community similarities, displayed much weaker time-dissimilarity relationships. Thus, their relationships were likely related to bacterial adaptation to the perturbed habitat, rather than the abundance and extent of community similarity.

## Comprehensive, yet versatile, coastal populations in response to different lignocellulose substrates

This study systematically investigated ligninolytic functional traits and taxonomic groups from a coastal wetland in response to various lignocellulose substrates. Lignin substrates are highly crossed-linked aromatic heteropolymers, with complex and varied structures. However, relatively few genes in metagenomes have been predicted to be involved in lignin degradation, as exemplified by 75 gene families from the Amazon River microbiome and 13 gene families from a North American forest microbiome participating in lignin degradation (2, 3). Our study extensively investigated 466 gene families involved in lignin metabolism, of which 394 gene families were identified in the samples (see Table S4 and S5 at https://doi.org/10.5281/zenodo.7659223). This not only improves the eco-enzymological understanding of microbiome mediated lignin transformation, but also challenges the current view of highly stable TerrOC in marine ecosystems. On one hand, our study demonstrated that *dyps* were the major ligninolytic genes in the coastal wetland, being a testament to the ubiquitous DyP activity of this niche (24) and is also in line with a previous report that lignin oxidation at coastal estuarine regions was mainly mediated by DyPs (3). However, this is divergent from lignin oxidation in North America forest regions, which preferred to utilize aryl alcohol oxidases (2). Such inconsistency indicates a difference in terrestrial carbon catabolism between soil and marine environments. Interestingly, the positive correlation between *dypB*s abundance and lignin content (Spearman's r = 0.78, $P < 0.001$) suggested that it could act as a lignin indicator to predict lignin content at coastal intertidal zones. Meanwhile, our study also examined a group of accessory enzyme genes (21, 35, 48), which have rarely been investigated in metagenome studies (2, 3, 13). Nineteen gene families, including reductases, oxidases, dehydrogenases, and ROS cycle enzymes, were revealed to potentially assist

ligninolytic oxidative enzymes (see Table S4 at https://doi.org/10.5281/zenodo.7659223). Additionally, genes involved in versatile lignin-derived aromatic compound pathways were identified in the coastal bacterial consortia. In contrast to the six gene families in aerobic β-ketoadipate pathway of a forest soil microbiome (2), and the 29 gene families in aerobic funneling pathways and 10 gene families in aerobic ring cleavage pathways from the Amazon River microbiome (3), our study revealed 371 gene families in lignin-derived aromatic compound pathways, including aerobic (175 gene families)/anaerobic (107 gene families) peripheral pathways and aerobic (57 gene families)/anaerobic (32 gene families) ring-cleavage pathways (see Table S4 at https://doi.org/10.5281/zenodo.7659223). Importantly, highly abundant genes in anaerobic pathways demonstrated anaerobic catabolism might be a major lignin degradation strategy for coastal consortia, which has rarely been reported previously (see Fig. S7C and Table S4 at https://doi.org/10.5281/zenodo.7659223). Also, in contrast to identified GH3 and GH10 gene families from the Amazon River microbiome (3), genes encoding free hemi-/cellulases and cellulosomes were detected in consortia at this niche, including GH3, GH5, GH9, and GH48 gene families for cellulose hydrolysis, GH5, GH10, GH11, GH16, and GH43 gene families for hemicellulose hydrolysis, as well as CBM2 and CBM3 as cellulose-binding protein families in cellulosomes. Together, a comprehensive set of genes associated with lignocellulose degradation existed within the coastal intertidal zone-derived consortia, revealing the unrecognized genomic potential for TerrOC degradation.

In contrast to previous reports which focused on the specialization of terrestrial taxa in the degradation of different lignocellulose components (cellulose, hemicellulose, and lignin) (2), specialization of coastal taxa was revealed in response to different lignocellulose substrates (Fig. 3). Moreover, substrate-dependent ligninolytic functional traits were observed between woody lignin and herbaceous lignin substrates, unlike the similar cellulosic degrading genes among different substrates (see Fig. S5 at https://doi.org/10.5281/zenodo.7659223). Thus, the clear linkages between potential decomposers and ligninolytic genes were established for woody and herbaceous lignin degradation. *Vibrio*, *Shewanella,* and *Arcobacter*, enriched by the woody lignin substrates, encoded *dypB*, gox, and/or other enzyme genes in ROS for lignin oxidation. This is also supported by our previous culture-dependent study, in which *Vibrio alginolyticus* and *Vibrio parahaemolyticus* strains, isolated from the *in situ* enrichment, exhibited DyP activity in the presence of Norway spruce lignin (24). In contrast, *Cohaesibacter* and *Labilibaculum*, enriched by herbaceous lignin, encoded *dypA* and/or auxiliary enzymes (e.g., *nuoE*, *nuoG,* and *gpx*) for lignin oxidation. Notably, the role of *Cohaesibacter* in TerrOC degradation has not been previously reported, revealing the unrecognized TerrOC degradation potential in the marine ecosystems. In addition, consortia enriched on the woody lignin substrates, were rich in a various of lignin-derived aromatic compound pathway genes. For instance, *Arcobacter*, *Vibrio*, *Desulfovibrio,* and *Pseudodesulfovibrio* were identified to contain genes involved in various anaerobic H-type lignin catabolism pathways, anaerobic benzoate pathway and anaerobic benzoyl-CoA ring cleavage pathways. *Arcobacter*, *Vibrio,* and *Shewanella* contained genes involved in aerobic G-, H-, S-type lignin degradation pathways, other aerobic lignin-derived aromatic compound (e.g., salicylic acid and 3-hydroxycinnamic acid) degradation pathways, and catechol-, PCA-ring cleavage pathways. However, fewer aromatic compound pathway genes, especially for those of anaerobic pathways, were identified in those members enriched on herbaceous substrate (e.g., *Labilibaculum*, *Cohaesibacter,* and *Bacteroides*) than that of consortia under woody substrates, possibly due to the lower lignin content of rice straw.

In all, our study identified 394 ligninolytic gene families and 154 hemi-/cellulosic gene families in the coastal intertidal wetland, uncovering great genomic potential in the coastal microbiome to degrade various lignocellulose substrates. Moreover, the links between ligninolytic functional traits and taxonomic groups of substrate-specific decomposers were revealed. It emphasizes the versatility present in microbiomes for lignin degradation, coinciding with the inherent heterogeneity of lignin. The pinpointed specific gene cocktails and specialized degraders could be further investigated

and developed as biocatalysts for the efficient and specific bioconversion of various lignocellulose substrates.

## Enhanced lignocellulose degradation by synergistic effect of lignin and hemi-/cellulose degradation

Cellulose and lignin degradation are mostly decoupled in aquatic ecosystems. For instance, cellulose and lignin degradation occurred in the different sections of the Amazon River (3). In contrast, our *in situ* enrichment experiment not only suggested that lignin and hemi-/cellulose were co-degraded, but also revealed higher lignin degradation rates under lignocellulose substrates than that of purified lignin substrate (Fig. 1; see also Table S2 at https://doi.org/10.5281/zenodo.7659223). Consistent with this, we observed enhanced bacterial abundance under lignocellulose substrates and highly interconnected co-occurrence networks of taxa and functional genes (see Fig. S1B and Fig. S8 at https://doi.org/10.5281/zenodo.7659223). These results supported a positive PE in the intertidal zone.

PE is thought to be a general phenomenon that occurs ubiquitously in various terrestrial and aquatic ecosystems, although its study in aquatic ecosystems, especially marine ecosystems, seriously lags behind (38, 62). However, a few studies have revealed its mechanism (62), especially in marine ecosystems (3). Here, we revealed the potential biological actors of PE at the level of taxa and functional genes under aerobic/anaerobic conditions. Moreover, the high interconnection relationship among these functional genes indicated close cooperation, suggesting the synergistic effect of lignin and hemi-/cellulose degradation. Surprisingly, genes encoding cellulosomes exhibited mostly negative connections with genes encoding anaerobic aromatic compound catabolic enzymes, which is distinct from the positive interactions between genes involved in free hemi-/cellulases and aerobic aromatic compound degradation. Future studies with dissolved organic matter (DOM)-microbe associations would provide insights into such specialization (63). It is interesting to note that the synergistic lignocellulose degradation probably occurred with the alternation of aerobic and anaerobic conditions (Fig. 5). Key aromatic metabolic branching points, e.g., catechol, benzoyl-CoA, and PCA, were pinpointed to connect anaerobic and aerobic aromatic compound metabolism. Such a key feature should contribute to lignocellulose degradation, as much more versatile strategies (aerobic or anaerobic pathways) could be employed in response to fluctuating micro-environments.

In conclusion, coastal intertidal wetlands are important transitional zones linking terrestrial and marine ecosystems. Our work represents a systematic effort to study the *in situ* bacterial metabolic potential for native TerrOC, which are inputs to these coastal systems. The results demonstrated how lignin/lignocellulosic substrates and exposure times affected long-term community assembly and indicated bacterial specialization in the degradation of various lignocellulose substrates. Furthermore, we identified the woody and herbaceous substrate-dependent functional genes for lignin degradation and revealed that the alternation of aerobic and anaerobic catabolism could facilitate PE in coastal intertidal wetlands, addressing several knowledge gaps in this area. Our work also is an important step to pave the road for future identification of anaerobic ligninolytic genes and characterization of complementary roles between coastal bacteria and fungi for lignocellulose degradation in marine ecosystems.

## MATERIALS AND METHODS

### *In situ* enrichment and sample collection

We used 16 cylindrical polytetrafluoroethylene (PTFE) incubators for *in situ* enrichment, each contained four chambers (10 cm height and 15 cm in radius) with 1 mm pores. The sterilized solid medium (10 g agar powder and 5 g phytagel in 1 L artificial seawater with a 1% (wt%) lignin/3% lignocellulose) were wrapped by 100 µM mesh enclosures,

were loaded in each chamber. As a result, each incubator contained one corresponding substrate. The substrates were commercially available Norway spruce lignin (Sigma-Aldrich, St. Louis, MO, USA, catalog# 370959), aspen wood powder, pine wood power, and rice straw powder. Aspen was collected from a forest farm in Liaocheng (Shandong Province, China). Pine was obtained from a forest farm in Zhenjiang (Jiangsu Province, China), while rice straw was obtained from a rice field in Zhoushan (Zhejiang Province, China). They were thoroughly washed, air dried (50°C), and then ground into pieces ≤5 mm by a hammer mill for *in situ* enrichment. Incubators were deployed in the intertidal zone (122°6′14.05 E, 29°56′48.90 N) of the south-eastern portion of Zhairuoshan Island, Zhejiang Province, China, and fixed to the beach by ropes. They were exposed during low tide, but submerged during high tide. Beginning in April 2019, each treatment with six replicates were enriched *in situ* for 3, 6, 12, and 18 months, respectively. The samples were collected in batches, kept on dry ice during transport, and stored at −80°C until further processing. The local temperature data were download from the website (http://tianqi.2345.com/wea_history/58477.htm). The pH, salinity, and DO concentration were measured every month, by pH meter (Satorius PB-10, Germany), salinity meter (ATAGO PAL-06S, Japan), and dissolved oxygen sensor (YSI Pro20, USA), respectively.

## DNA extraction and high-throughput sequencing

Approximately 20 g of each sample was collected for DNA extraction. DNA was extracted by the CTAB extraction method as previously described (64). As result, ~7–117 µg DNA was generated and preserved at −80°C for subsequent 16S rRNA amplicon sequencing and shotgun metagenomic sequencing.

To investigate the enriched bacterial taxonomic compositions, all DNA samples (*n* = 96) were used for 16S rRNA amplicon sequencing (Fig. 1A). The V4 region of the bacterial 16S rRNA gene with unique barcodes was amplified using the primer set 515F/806R (515F: 5'-GTGCCAGCMGCCGCGGTAA-3' and 806R: 5'-GGACTACHVGGGTWTCTAAT-3') (65). The amplified products were pooled in equimolar concentrations, and loaded onto an Illumina PE250 platform at Novogene Co., Ltd., Beijing, China, for high-throughput sequencing (66). A total of 6,259,640 reads were collected, with an average of 65,205 reads per sample and a standard deviation of 10,058.

To explore different compositional patterns for functional genes among the various lignocellulose enrichments, DNA from samples with 6 months exposure (*n* = 12) were used for metagenomic DNA sequencing on an Illumina PE150 platform at Novogene Co., Ltd., Beijing, China, with 150 bp paired-end sequencing (Fig. 1A). Moreover, to construct functional genes co-occurrence network, another six samples, which were enriched in a previous *in situ* enrichment (6 months) study (24), were also used for metagenomic DNA sequencing. The six samples were enriched from April 2018 to October 2018 at the same location, with 3% rice straw powder, corn straw powder, and wheat straw powder as the carbon source, respectively (24). As a result, the metagenomic DNA data sets targeting 18 samples were obtained. A total of 1,499,127,964 reads were collected, with an average of 124,927,330 reads per sample and a standard deviation of 13,119,038. All sequencing data have been deposited in the NCBI SRA database under the accession number PRJNA836095.

## High-throughput sequencing data analysis

For the 16S rRNA gene sequencing data, all subsequent sequence processing of 16S sequencing paired-end reads were performed using the "DADA2 R" package (v1.22.0) and associated pipeline, for primer removal, denoising, filtering, merging, and chimera removal from these sequences and to generate ASVs (67). To equalize sequencing depth, sequencing results were subsequently rarefied to 51,498 reads, based on the sample with the lowest sequence number. Taxonomic identification of ASVs, at a 100% identity, was assigned against the Ribosomal Database Project (RDP) 16S rRNA gene training set 18 using the Bayesian classifier at 80% confidence level (68). Linear discriminant analysis (LDA) Effect size (LEfSe) was performed to identify the significant biomarker taxa using

the online pipeline (http://huttenhower.sph.harvard.edu/galaxy), with LDA score = 3.5 and $P$ < 0.05. Meanwhile, the Spearman correlation was calculated to investigate the relationships between substrate compositions (cellulose, hemicellulose, and lignin) and ASVs. This was carried out using the "psych" package (v2.2.9) (69) and visualized by the "pheatamp" package (v1.0.12) (70) in R (v4.1.3).

For metagenomic sequencing data, Illumina adapters and low-quality sequences were removed from metagenomes using Trimmomatic (v0.36) with the parameters: ILLUMINACLIP:TruSeq3-SE:2:30:10 LEADING:3 TRAILING:3 SLIDINGWINDOW:4:15 MINLEN:36 (71). The quality of the reads was evaluated with FASTQC (v0.11.7) (72). Next, as previously described (73), read-based analyses rather than metagenomic assembly were carried out to get a more representative sequence set for the microbial communities involved in lignocellulose degradation. Genes involved in hemi-/cellulose hydrolysis (161 gene families) and lignin degradation (466 gene families) were collected based on literature searches (see Table S5 at https://doi.org/10.5281/zenodo.7659223) (21, 53, 74–77). For genes involved in hemi-/cellulose hydrolysis, representative genes (and their predicted amino acid sequences) were annotated by searching against the carbohydrate-active enzymes (CAZymes) database (release 09242021) (78) via the Blastx algorithm implemented in Diamond (v0.9.25), with identity ≥70% and e-value ≤ 1 $e^{-5}$ (79). Similarly, representative genes, involved in lignin degradation, were annotated by searching against eggNOG (80), arCOG (81), KEGG (82), and COG (83) database with identity ≥70% and e-value ≤ 1 $e^{-5}$ (79, 84).

Three different complementary nonparametric analyses for metagenomic data were used: nonparametric multivariate analysis of variance (Adonis) using distance matrices (85), analysis of similarity (ANOSIM) (86), and multiple response permutation procedure (MRPP) (87). The Bray–Curtis distances were used to calculate the dissimilarities between consortia enriched by woody and herbaceous substrates (88). All three procedures (ANOSIM, Adonis, and MRPP) were performed with the "vegan" package (v2.5.7) in R software v4.1.3.

Taxonomic classification was performed based on the DIAMOND output files. Metagenomic sequences mapped to targeting gene families (cellulose/hemicellulose/lignin) were extracted by the seqtk program, and the tool Kraken 2 was used for taxonomic assignment of these extracted sequences, as previously described (89). A local standard Kraken 2 database was built for taxonomic assignment.

## Consortium diversity and variance analysis

For ASV community compositions, several indices of alpha diversity were calculated, including Shannon, Richness and Evenness. Comparisons of the alpha diversity between two groups was performed with Wilcoxon rank-sum test. For beta diversity, NMDS and permutational multivariate analysis of variance (PERMANOVA) were performed based on Bray–Curtis distance. These analyses were conducted with the "vegan" package (v2.5.7) in R (v4.1.3) (90).

For the time dependent dissimilarity relationship, pairwise temporal distances between samples were assessed against Bray–Curtis dissimilarities by the "ggplot2" package (v3.4.0) in R (91). Spearman's rank correlations between Bray–Curtis dissimilarities and temporal distances was calculated. ASVs belonging to major bacterial taxa (>1%) were also examined their receptive temporal-decay relationships.

Mantel tests were conducted to further verify the contributions of substrates, exposure time and environmental factors involved in exposure time, e.g., temperature, salinity, pH, and DO, to bacterial community variations (90).

## Null model

Phylogenetic bin-based null model analysis (iCAMP), with 16S rRNA gene sequencing data, was applied to quantify the contribution of various ecological processes to community assembly (92). It describes assembly processes, e.g., heterogeneous/homogeneous selection, dispersal limitation, homogenizing dispersal, and drift, based

on a quantitative framework (92). Briefly, the observed taxa were first divided into different phylogenetic bins based on their phylogenetic relationships. The process governing each bin was then identified based on null model analysis of phylogenetic diversity using beta Net Relatedness Index (βNRI) and taxonomic β-diversities using modified Raup–Crick metric (RC). For each bin, the fraction of pairwise comparisons with βNRI < −1.96 and > +1.96 were considered as the percentages of homogeneous and heterogeneous selection, respectively (93). Next, RC was used to partition the remaining pairwise comparisons with |βNRI| ≤ 1.96: the fraction of pairwise comparisons with RC < −0.95 and > +0.95 are treated as the percentages of homogenizing dispersal and dispersal limitation (94), and the remaining fraction, with |βNRI| ≤ 1.96 and |RC| ≤ 0.95, represented the percentages of drift (94). The above analysis was repeated for every bin, and then the fractions of individual processes across all bins were further weighted by the relative abundance of each bin and summarized to estimate the relative importance of individual processes at the whole community level (95).

## Network analysis

Co-occurrence networks were constructed based on Spearman's rank correlations, due to its more sophisticated network inference algorithms (e.g., inference in the presence of noise) (96, 97). All samples ($n$ = 96) with 16S rRNA gene sequencing were used to construct the taxon co-occurrence network. Only ASVs with relative abundance >0.01% across all samples and occurring in at least 75% of all samples were selected for the network construction, and the correlations with thresholds value Spearman's | r | > 0.6 and $P$ value < 0.05 were displayed, as previously described (98).

Metagenome samples ($n$ = 18) were used to construct the functional genes co-occurrence network. Similarly, genes occurring in at ≥ 75% of all samples were chosen, and the correlations with thresholds value Spearman's | r | > 0.7 and $P$ value < 0.05 were displayed (99). Co-occurrence networks were visualized by Cytoscape software (100).

## Quantitative PCR

The abundance of total bacteria in samples after 6 months exposure was measured by the quantitative PCR (qPCR) using the primer set 341F/519R (341F: 5'-CCTACGGGW-GGCWGCA-3' and 519R: 5'-TTACCGCGGCKGCTG-3'), with a qPCR protocol that has been previously described (101, 102). For each sample, triplicate amplifications were conducted. Standard curves were generated from tenfold serially diluted linear plasmids containing a single copy of the bacterial 16S rRNA gene. All experiments were performed in biological triplicate and technical duplicate.

## Substrate measurement and analysis

Approximately 8 g of each sample was mechanical disrupted, agar melted and then filtered through nonwoven fabrics (30 g/m$^2$, 30 × 30 cm). The filtered residue was dried after washing off for further analysis. Substrate compositions (cellulose, hemicellulose. and lignin) were determined by the Laboratory Analysis Protocol (LAP) of the National Renewable Energy Laboratory (NREL), Golden, CO, USA. Briefly, 95% ethanol was used to extract samples for 8 h, using the Soxhlet method. Subsequently, 72% H$_2$SO$_4$ was added in samples at 30℃ for 1 h. Next, the slurry was diluted to 4% by adding milliQ pure water and incubated in an autoclave for 1 h at 121℃. The residual was filtered by filtering crucibles (25 mL, porcelain, medium porosity) and dried in Muffle furnace at 575℃ for at least 4 h. The acid-insoluble lignin fraction was measured by the difference of weight loss, while acid-soluble lignin content (ASL) was detected by UV-Vis spectroscopy at 320 nm. A blank sample without enrichment was used as a reference control.

Subsequently, a high-performance liquid chromatograph (HPLC, Agilent 1260 Infinity II) equipped with a refractive index detector (RID) was employed to measure concentrations of cellulose and hemicellulose. HPLC analysis was conducted using an Agilent Hi-Plex H analysis column, with a flow rate of 0.4 mL/min. Cellulose and hemicellulose

standard samples (xylose, glucose, and arabinose), which were purchased from the Solarbio company, were also analyzed by HPLC according to the LAP of the NREL. All experiments were performed in biological triplicate and technical duplicate. Differences between groups were evaluated using Wilcoxon rank-sum test. Those with $P$ values of < 0.05 were considered significant.

## Quantitative $^{31}$P nuclear magnetic resonance analysis of lignin functional groups

To quantify the lignin functional groups in each substrate, $^{31}$P nuclear magnetic resonance (NMR) analysis was performed. Lignin was extracted from the lignocellulose substrate (e.g., aspen, pine, and rice straw) as previously described (103, 104). Briefly, lignocellulose was milled into powders (40–60 mesh), dried at 60℃ and dewaxed with ethanol/toluene (1:2, vol/vol) in a Soxhlet extractor for 8 h. Subsequently, the extractive-free sample was air-dried and ball-milled using a ball mill at 60 Hz with zirconium dioxide vessels (50 mL) containing ZrO2 ball bearings (20 mm × 2) for 2 h (2 min grinding and 2 min break). Three grams of ball-milled sample was then hydrolyzed by Cellic CTec3 (0.1 mL/g biomass) and Hemicellulase (0.1 mL/g biomass) in 100 mL 50 mM citrate buffer (pH 4.8) at 50℃, 150 rpm for 48 h. The supernatants were removed by centrifugation (6,500 g, 10 min). The solid residues were washed with 50 mM sodium citrate buffer and milliQ pure water, and then enzymatically hydrolyzed again using the above procedure. The lignin was extracted with 96% dioxane at ambient temperature for 48 h and recovered by a rotary evaporator (55℃ for 0.5 h). Finally, lignin was lyophilized by lyophilizer (Scientz-18N, Ningbo Scientz Biotechnology Co., Ltd, Ningbo, China) at −80℃ and ≤5 pa mPa for a minimum of 48 h for $^{31}$P NMR analysis.

$^{31}$P NMR spectra were acquired using a Bruker Avance 600 MHz NMR spectrometer. The lignin sample (25 mg) was dissolved in a solvent of pyridine/CDCl$_3$ (1.6:1.0 vol/vol, 550 µL) containing chromium acetylacetonate (CCA, relaxation agent) and endo-N-hydroxy-5-norbornene-2,3-dicarboximide (NHND, internal standard), shaking until the lignin was dissolved completely, and then derivatized with 2-chloro-4,4,5,5-tetra-methyl-1,3,2-dioxaphospholane (TMDP) (44, 105). The spectrum was acquired using an inverse-gated decoupling pulse sequence (Waltz-16), a 90° pulse, and a 25 s pulse delay. One hundred twenty-eight scans were accumulated for each sample. NMR data were processed using the TopSpin 4.0.8 software (Bruker BioSpin) software packages. All experiments were performed in biological triplicate.

## ACKNOWLEDGMENTS

We would like to thank Haiyan Sui from Shandong University core facilities for Life and Environmental Sciences for their help and guidance in the NMR assay.

This work was supported by the National Key Research and Development Project of China (2018YFA0605800), National Natural Science Foundation of China (91951116 and 92051110) and National Key Research and Development Project of China (2020YFA0607600 and 2019YFA0606704).

L.L. and Q.T. conceived and designed the study. Q.P. extracted DNA, measured the substrates, analyzed the data and prepared the figures and tables. X.W. performed in situ enrichment experiment. Y.Z. measured the environmental factors. Q.T. and J.C. assisted the metagenome data analysis. L.L., Q.P., J.Z., and N.J wrote the manuscript. All authors approved the final manuscript.

The authors declare that they have no competing interests.

## AUTHOR AFFILIATIONS

[1]Institute of Marine Science and Technology, Shandong University, Qingdao, China
[2]Key Laboratory of Applied Marine Biotechnology, Ministry of Education, Ningbo University, Ningbo, China

[3]State Key Laboratory of Marine Environmental Science and College of Ocean and Earth Sciences, Xiamen University, Xiamen, China

[4]Joint Lab for Ocean Research and Education at Shandong University, Xiamen University and Dalhousie University, Qingdao, China

[5]Institute for Environmental Genomics, University of Oklahoma, Norman, Oklahoma, USA

[6]Department of Microbiology and Plant Biology, University of Oklahoma, Norman, Oklahoma, USA

[7]School of Civil Engineering and Environmental Sciences, University of Oklahoma, Norman, Oklahoma, USA

[8]School of Computer Science, University of Oklahoma, Norman, Oklahoma, USA

## AUTHOR ORCIDs

Qiannan Peng http://orcid.org/0000-0002-7839-9846

Lu Lin http://orcid.org/0000-0002-9663-6382

Qichao Tu http://orcid.org/0000-0002-3245-7545

## FUNDING

| Funder | Grant(s) | Author(s) |
| --- | --- | --- |
| MOST | National Key Research and Development Program of China (NKPs) | 2018YFA0605800 | Lu Lin |
| MOST | National Natural Science Foundation of China (NSFC) | 91951116 | Lu Lin |
| MOST | National Key Research and Development Program of China (NKPs) | 2019YFA0606704 | Lu Lin |
| MOST | National Key Research and Development Program of China (NKPs) | 2020YFA0607600 | Qichao Tu |
| MOST | National Natural Science Foundation of China (NSFC) | 92051110 | Qichao Tu |

## AUTHOR CONTRIBUTIONS

Qiannan Peng, Data curation | Lu Lin, Funding acquisition, Project administration, Supervision, Writing – original draft, Writing – review and editing | Qichao Tu, Funding acquisition, Methodology, Project administration | Xiaopeng Wang, Data curation | Yueyue Zhou, Data curation | Jiyu Chen, Data curation | Nianzhi Jiao, Conceptualization, Supervision | Jizhong Zhou, Conceptualization, Supervision, Writing – review and editing

## DATA AVAILABILITY

All sequencing data have been deposited in the NCBI SRA database under the accession number PRJNA836095. The analyzed 16S rRNA data and metagenome data are detailed in Supplementary Tables S3 and S4, respectively. The supplementary material is available at Zenodo.

## ADDITIONAL FILES

The following material is available online.

Open Peer Review

**PEER REVIEW HISTORY (review-history.pdf).** An accounting of the reviewer comments and feedback.

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
