## [Reviewer comments · mSystems]

Unraveling the roles of coastal bacterial consortia in degradation of various lignocellulosic substrates

Qiannan Peng, Lu Lin, Qichao Tu, Xiaopeng Wang, Yueyue Zhou, Jiyu Chen, Nianzhi Jiao, and Jizhong Zhou

Corresponding Author(s): Lu Lin, Shandong University

Review Timeline:

Submission Date:	December 20, 2022
Editorial Decision:	February 8, 2023
Revision Received:	February 21, 2023
Editorial Decision:	March 27, 2023
Revision Received:	April 6, 2023
Accepted:	May 12, 2023

Editor: Zarath Summers

Reviewer(s): Disclosure of reviewer identity is with reference to reviewer comments included in decision letter(s). The following individuals involved in review of your submission have agreed to reveal their identity: Shi Huang (Reviewer #1)

Transaction Report:

DOI: <https://doi.org/10.1128/msystems.01283-22>

February 8, 2023

Prof. Lu Lin
Shandong University
Qingdao
China

Re: mSystems01283-22 (Unraveling the roles of coastal bacterial consortia in degradation of various lignocellulose substrates)

Dear Prof. Lu Lin:

Thank you for submitting your manuscript to mSystems. We have completed our review and I am pleased to inform you that, in principle, we expect to accept it for publication in mSystems. However, acceptance will not be final until you have adequately addressed the reviewer comments.

Please ensure that you have 10 or fewer supplementary figures. Some will need to be removed to proceed.

Preparing Revision Guidelines

Sincerely,

Zarath Summers

Editor, mSystems

Journals Department
Reviewer comments:

Reviewer #1 (Comments for the Author):

The authors investigated and characterized microbial populations and functional genes attributed to different lignocellulose (hardwood, softwood, and herbaceous) and lignin substrates using in situ lignocellulose enrichment experiments coupled with 16S rRNA amplicon and shotgun metagenomics sequencing methods. The different impacts of substrates and exposure times on long-term bacterial communities and the links of ligninolytic functional traits and taxonomic groups of substrate-specific populations were revealed. Overall, the paper has been well written. However, a few technical concerns need to be addressed for publication.

1. Given the relatively complex study design, I suggest authors to add a sub-figure showing how they collected samples including sample size and what techniques have been applied to the samples from different habitats.
2. Only the metagenomics method was used in this study, please be aware that DNA from those dead bacterial cells can be also detected. Therefore, although this study provided many DNA-level signals from the bioinformatic analysis, it remains unknown about the gene expression or microbiome activities associated with lignin degradation. On top of this study, more validation work with meta-transcriptome or culturomics would be needed in the future.
3. Any particular rationale for selecting the threshold for spearman correlation in the co-occurrence network? I saw you used different thresholds for 16S and whole metagenomics data.
4. Typo or grammatical errors can be found sporadically. Please check the language issues thoroughly in the manuscript.
5. Line 787, Wilcoxon rank sum test or Wilcoxon signed-rank test? Which method was used should be clearly stated as you can do the mean comparison of certain metrics between time points in this study.
6. The reference format needs to be carefully checked. For example, the page number of references needs to be complemented, e.g., lines 990, 965, 855, 859, 868, 918, 938, and 948.

Reviewer #2 (Comments for the Author):

Peng et al examined bacterial communities associated with the degradation of lignocellulose substrates via an in-situ enrichment experiments. The bacterial communities were explored by both amplicons and shotgun metagenomic sequencing, and the relevant functional genes associated with lignocellulose and lignin substrates were examined. They included the commercially available lignocellulose of hardwood, softwood and herbaceous. They found the consortia enriched on woody lignocellulose showed higher diversity than those on the herbaceous substrate. A time-dissimilarity pattern with increased alpha diversity over time was observed. Their findings are challenging current opinions of lignin stability in marine ecosystems by revealing a comprehensive set of genes associated with lignin degradation potential, containing 23 gene families involved in lignin depolymerization, and 371 gene families involved in aerobic/anaerobic lignin-derived aromatic compound pathways. Some results may be not surprising but are interesting considering this rarely studied transitional zone. I have only some minor comments for better clarity and hope the authors found them helpful.

More details could be provided in the experimental setups. For instance, the authors included samples from another in-situ experiments (but shorter durations). I would expect more information provided for such inclusions. Are these two experiments started at the same time, and how about the seasons covered for these two experiments?

Environmental conditions are not well examined, which is necessary for an in-situ experiments. For instance, how about the importance of temperature and other local environments underlying the community composition, functional compositions, and ecological processes? Better to incorporate important results available in supplementary materials into the main figures?

L29. Change to "there are only few studies in"?

L43. Traditional view?

L46. Better introduction of priming effects here? The readers may not expect the appearance of priming effects at the end of abstract.

L48. Change to indicates. Same for other locations in the following sections.

L111. What is DO?

L178. Wilcoxon text?

L188. What is NMDS analysis?

L225-227. Rewrite this sentence for better understanding.

L227. Change to suggest.

L228. What is null model analysis? What are stochastic or deterministic processes? Proper citations or better explanations should be provided.

L263. What kind of three tests?

L335. What coefficient? Rho or r?

L336. Change to "was constructed with 288 nodes".

L360. Rewrite this sentence? I could not understand the "no significantly higher correlation".

L392. It is interesting to reveal the network specialization. Would this be linked to other specialization analysis such as Hu et al (Nat Commun 13:3600).

L395. A brief introduction of this synergistic models are needed here.

L402. Too strong to say "should work together"?

L430. Any relevant references could be cited for this cross-feeding cooperation as described here?

L497. Sharp contrast? Better writing?

L578. It is a bit lost here to read something relevant to "Amazon River". Better reorganization of this paragraph is expected.

L605. Rewrite as "zones to link terrestrial and marine ecosystems"?

L644-647. I am not sure so much DNA could be yield with 20 mg samples. Better double check.

L648-655. Provide proper references.

L656-664. More information or reasonings are needed for inclusion of another experiments.

L695. Provide proper references.

L711. Should be richness, evenness.

L720. Please check whether Spearman or Pearson analysis were used in the Results section, and proper results should be reported.

L726. More relevant references should be provided.

L758-817. Please check the results relevant to qPCR and NMR are well reported in the Result section.

L768. What kind of filter paper?

Feb. 20, 2023
Dr. Zarath M. Summers, Editor
mSystems

Dear Editor Zarath M. Summers

Thank you very much for giving us the opportunity to improve our manuscript (mSystems01283-22) by responding to the extremely valuable reviewer/editor comments! Here, we are submitting a revised version, with item-by-item responses (regular fonts) to reviewers' suggestions (*italicized* fonts) below.

Comments from editor:

1) *Please ensure that you have 10 or fewer supplementary figures. Some will need to be removed to proceed.*

Response: Yes. We have revised this section and kept the 10 supplementary figures in the revised version. Please refer to the revised supplementary materials.

Comments from reviewers:

Reviewer comments:

Reviewer #1 (Comments for the Author):

2) *The authors investigated and characterized microbial populations and functional genes attributed to different lignocellulose (hardwood, softwood, and herbaceous) and lignin substrates using in situ lignocellulose enrichment experiments coupled with 16S rRNA amplicon and shotgun metagenomics sequencing methods. The different impacts of substrates and exposure times on long-term bacterial communities and the links of ligninolytic functional traits and taxonomic groups of substrate-specific populations were revealed. Overall, the paper has been well written. However, a few technical concerns need to be addressed for publication.*

Response: We very much thank the reviewer for their positive comments!

3) *Given the relatively complex study design, I suggest authors to add a sub-figure showing how they collected samples including sample size and what techniques have been applied to the samples from different habitats.*

Response: That's an excellent suggestion! We have added the figure, as Figure 1a, in the revised version to present the overview of experimental design. Now it reads (*see Figure 1* and its legend).

Figure 1 *in situ* lignocellulose enrichment experiment. (A) Schematic representation of the experiment design. A: aspen, P: pine, NS: Norway spruce lignin, R: rice straw. (B) Lignocellulosic composition of each substrate. (C-F) Percentage of consumed lignin, cellulose and hemicellulose in each substrate by enriched bacteria consortia over time. 3M-18M: different amounts of exposure time, M indicating months. Significant differences between samples are indicated by asterisks (*, $p < 0.05$, **, $p < 0.01$).

4) *Only the metagenomics method was used in this study, please be aware that DNA from those dead bacterial cells can be also detected. Therefore, although this study provided many DNA-level signals from the bioinformatic analysis, it remains unknown about the gene expression or microbiome activities associated with lignin degradation. On top of this study, more validation work with meta-transcriptome or culturomics would be needed in the future.*

Response: We acknowledge this important comment by the reviewer! We also have noticed this point and are performing corresponding experiments (RNA data and culture-omics) to investigate the lignocellulose catabolic processes of coastal bacterial communities. A manuscript is now under preparation specifically for this issue.

5) *Any particular rationale for selecting the threshold for spearman correlation in the co-occurrence network? I saw you used different thresholds for 16S and whole metagenomics data.*

Response: Based on the literatures and empiric values, we selected the threshold of spearman correlation to maintain appropriate information in the network. We have revised it to ensure accurate communication. The text now reads (*see Page 33, Line 798-807*):

“Only ASVs with relative abundance $> 0.01\%$ across all samples and occurring in at least 75% of all samples were selected for the network construction, and the correlations with thresholds value Spearman's $|r| > 0.6$ and p value < 0.05 were displayed, as previously described (1).

Metagenome samples ($n=18$) were used to construct functional genes co-occurrence network. Similarly, genes occurring in at $\geq 75\%$ of all samples were chosen, and the correlations with thresholds value Spearman's $|r| > 0.7$ and p value < 0.05 were displayed (2). Co-occurrence networks were visualized by Cytoscape software (3).”

6) *Typo or grammatical errors can be found sporadically. Please check the language issues thoroughly in the manuscript.*

Response: Yes. We have thoroughly proof-read and improved English usage (including correcting non-standard expressions and grammatical errors) across the main-text and supplementary materials.

7) *Line 787, Wilcoxon rank sum test or Wilcoxon signed-rank test? Which method was used should be clearly stated as you can do the mean comparison of certain metrics between time points in this study.*

Response: Wilcoxon rank sum test was used in this study. We have revised it to ensure accurate understanding. The text now reads (*see Page 34-35, Line 838-840*):

“Differences between groups were evaluated using Wilcoxon rank sum test. Those with p values of < 0.05 are considered significant.”

8) *The reference format needs to be carefully checked. For example, the page number of references needs to be complemented, e.g., lines 990, 965, 855, 859, 868, 918, 938, and 948.*

Response: Yes. We have corrected it accordingly. Please refer to **Page 37-44, Line 889-1213** in the revised version.

Reviewer #2 (Comments for the Author):

9) *Peng et al examined bacterial communities associated with the degradation of lignocellulose substrates via an in-situ enrichment experiments. The bacterial communities were explored by both amplicons and shotgun metagenomic sequencing, and the relevant functional genes associated with lignocellulose and lignin substrates were examined. They included the commercially available lignocellulose of hardwood, softwood and herbaceous. They found the consortia enriched on woody lignocellulose showed higher diversity than those on the herbaceous substrate. A time-dissimilarity pattern with increased alpha diversity over time was observed. Their findings are challenging current opinions of lignin stability in marine ecosystems by revealing a comprehensive set of genes associated with lignin degradation potential, containing 23 gene families involved in lignin depolymerization, and 371 gene families involved in aerobic/anaerobic lignin-derived aromatic compound pathways. Some results may be not surprising but are interesting considering this rarely studied transitional zone. I have only some minor comments for better clarity and hope the authors found them helpful.*

Response: We thank the reviewer very much for these positive comments on this work.

10) *More details could be provided in the experimental setups. For instance, the authors included samples from another in-situ experiments (but shorter durations). I would expect more information provided for such inclusions. Are these two experiments started at the same time, and how about the seasons covered for these two experiments?*

Response: Yes. The two experiments both covered spring and summer. We have added the detailed information to ensure accurate understanding. The text now reads

(*see Page 28, Line 678-681*): “Beginning in April 2019, each treatment with six replicates were enriched *in situ* for 3-, 6-, 12- and 18-month respectively. The samples were collected in batches, kept at dry ice during transport, and stored at -80°C until further processing.”

(*see Page 29, Line 700-712*): “To explore different compositional patterns for functional genes among various lignocellulose enrichments, the DNA samples (n=12) with 6-month exposure were used for metagenomic DNA sequencing on an Illumina PE150 platform at Novogene Co., Ltd., Beijing, China, with 150 bp paired-end sequencing (Fig. 1A). Moreover, to construct functional genes co-occurrence network, another 6 samples, which were enriched in a previous *in situ* enrichment (6-month) study (4), were also used for metagenomic DNA sequencing. The 6 samples were enriched from April 2018 to October 2018 at the same location, with 3% rice straw powder, corn straw powder and wheat straw powder as the carbon source, respectively (4). As a result, the metagenomic DNA data sets targeting 18 samples were obtained. A total of 1,499,127,964 reads were collected, with an average of 124,927,330 reads per sample and a standard deviation of 13,119,038. All sequencing data have been deposited in the NCBI SRA database under the accession number PRJNA836095.”

11) *Environmental conditions are not well examined, which is necessary for an in-situ experiments. For instance, how about the importance of temperature and other local environments underlying the community composition, functional compositions, and*

ecological processes? Better to incorporate important results available in supplementary materials into the main figures?

Response: Yes. We have showed the result in the Figure 2 to sufficiently describe the influence of environmental factors for the community composition and ecological processes. Please refer to **Page 10-11, Line 227-253** in the revised version.

The **Figure 2** and its legend now read:

Figure 2 The diversity and assembly mechanism of the enriched bacterial communities. (A) Shannon diversity of the bacterial taxonomic community over enrichment time. Significant differences between sampling areas are indicated by asterisks (*, $p < 0.05$, **, $p < 0.01$). (B) A non-metric multidimensional scaling (NMDS) profile of the 96 bacterial taxonomic communities based on the Bray-Curtis dissimilarities. Ellipses depict the 95% confidence interval grouping effects of exposure time, represented by different colors. Stress is included in the upper right corner. (C) The association of environmental factors and bacterial taxonomic communities. The compositional variations for taxonomic groups were analyzed. Mantel's r was calculated for association strength. (D) Dynamic changes of stochasticity and determinism during the succession of bacterial taxonomic communities.

12) L29. Change to "there are only few studies in"?

Response: Yes. We have revised it accordingly.

13) L43. Traditional view?

Response: Yes. We have revised it accordingly.

14) L46. Better introduction of priming effects here? The readers may not expect the appearance of priming effects at the end of abstract.

Response: Yes. We have revised it to ensure understanding. The text now reads (*see Page 2, Line 45-49*):

“Importantly, we not only observed synergistic degradation of lignin and hemi-/cellulose, but also pinpointed the potential biological actors at the levels of taxa and functional genes, which indicates that the alternation of aerobic and anaerobic catabolism could facilitate lignocellulose degradation.”

15) L48. *Change to indicates. Same for other locations in the following sections.*

Response: Yes. We have revised it at here and across the manuscript.

16) L111. *What is DO?*

Response: It's dissolved oxygen. We have revised it to ensure understanding. The text now reads (*see Page 5, Line 110-112*):

“Meanwhile, exposure time is the comprehensive factor, including multiple environmental factors (e.g., temperature, pH, salinity and dissolved oxygen (DO)).”

17) L178. *Wilcoxon text?*

Response: Here, we used Wilcoxon rank sum test to compare richness of consortia at each time point. We have revised it to ensure understanding. The text now reads (*see Page 8, Line 178-181*):

“On one hand, richness (the number of unique amplicon sequence variants (ASV)) significantly increased ~3.5-fold from month 3-month to 18-month (Wilcoxon rank sum test, see Fig. S1C at <https://doi.org/10.5281/zenodo.7659223>).”

18) L188. *What is NMDS analysis?*

Response: It's non-metric multidimensional scaling analysis for beta diversity. We have revised it to ensure understanding. The text now reads (*see Page 9, Line 193-195*):

“Non-metric multidimensional scaling (NMDS) analysis showed a temporal separation of consortia across all substrates (Fig. 2B).”

19) L225-227. *Rewrite this sentence for better understanding.*

Response: Yes. We have rewritten it to improve the readability. The text now reads (*see Page 10, Line 233-237*):

“Hence, Mantel test was also used to assess the effects of environmental factors, including pH, temperature, salinity and DO. These factors all significantly associated with the compositional variations of bacterial consortia ($p = 0.001$, Fig. 2C). Among them, temperature was the most influencing factor associating with the taxonomic composition, followed by pH, salinity and DO, respectively (Fig. 2C).”

20) L227. *Change to suggest.*

Response: Yes. We have revised it accordingly. Please refer to **Page 10, Line 237-238** in the revised version.

21) L228. *What is null model analysis? What are stochastic or deterministic processes? Proper citations or better explanations should be provided.*

Response: Yes. We have revised it to improve the readability. The text now reads (*see Page 10-11, Line 238-245*):

“To verify this, null model analysis was performed to investigate the relative importance of deterministic and stochastic processes in the assembly of consortia. It

showed that homogenous selection (one of deterministic processes) was the most important process, accounting for 53.9%-59.2% of the community variation across all communities, followed by stochastic processes, such as drift (18.1%-30.8%) and dispersal limitation (15.0%-22.1%, see Fig. S4A at <https://doi.org/10.5281/zenodo.7659223>).

In addition, relevant references have been cited in the “Methods” section for the description of null mode. Please refer to **Page 32-33, Line 774-793** in the revised version.

22) L263. *What kind of three tests?*

Response: They are permutational multivariate analysis of variance (Adonis), analysis of similarity (ANOSIM) and multiple response permutation procedure (MRPP). We have inserted a detailed explanation to ensure understanding. The text now reads (*see Page 12, Line 277-281*).

“Interestingly, the compositions of gene families encoding cellulases among the four substrates were similar ($p > 0.05$), as indicated by the three complementary nonparametric tests, permutational multivariate analysis of variance (Adonis), analysis of similarity (ANOSIM) and multiple response permutation procedure (MRPP) (see Fig. S5 at <https://doi.org/10.5281/zenodo.7659223>).

23) L335. *What coefficient? Rho or r?*

Response: It’s Spearman’s r . We have revised it. The text now reads (*see Page 15, Line 363-365*):

“As a result, a network was constructed with 288 nodes and 5699 edges (Spearman’s $|r| > 0.6$ and p value < 0.05 , see Fig. S8B at <https://doi.org/10.5281/zenodo.7659223>).

24) L336. *Change to "was constructed with 288 nodes".*

Response: Yes. We have revised it accordingly. Please refer to our response to Review-Comment No. 23 above.

25) L360. *Rewrite this sentence? I could not understand the "no significantly higher correlation".*

Response: Yes. We have revised it to improve the readability. The text now reads (*see Page 16-17, Line 391-394*):

“Interestingly, links among aerobic ligninolytic oxidative genes weren’t observed, indicating they were likely to independently depolymerize lignin, presenting an aggregation mode for lignin depolymerization.”

26) L392. *It is interesting to reveal the network specialization. Would this be linked to other specialization analysis such as Hu et al (Nat Commun 13:3600).*

Response: That’s an excellent suggestion! We have cited this literature and discussed to further reveal such specialization with other complement methods in the next step. The text now reads (*see Page 26-27, Line 637-642*):

“Surprisingly, genes encoding cellulosomes exhibited mostly negative connections with gene encoding anaerobic aromatic compound catabolic enzymes, which is distinct with the positive interactions between genes involved in free hemi-/cellulases and aerobic aromatic compound degradation. Future studies with dissolved organic

matter (DOM)-microbe associations would provide insights into such specialization (5).”

27) L395. *A brief introduction of this synergistic models are needed here.*

Response: Yes. We have revised it to improve the readability. The text now reads (*see Page 18, Line 429-431*):

“Finally, integrating all of the above-stated information, a synergistic model for lignocellulosic degradation was proposed, which identified the potential taxonomic groups and functional genes (Fig. 5).”

28) L402. *Too strong to say "should work together"?*

Response: Yes. We have revised the statement. The text now reads (*see Page 18, Line 436-439*):

“The interconnection at taxon and gene levels suggest that these potential degraders could work together to efficiently hydrolyze hemi-/cellulose via the aggregation and division of labor strategies (e.g., aerobic and anaerobic hydrolysis).”

29) L430. *Any relevant references could be cited for this cross-feeding cooperation as described here?*

Response: Yes. We have cited the literatures at here to ensure understanding. The text now reads (*see Page 19, Line 465-467*):

“Such cross-feeding cooperation (6, 7) would stimulate aromatic compound metabolism under fluctuating micro-environments.”

30) L497. *Sharp contrast? Better writing?*

Response: Yes. We have revised the statement. The text now reads (*see Page 22, Line 534-539*):

“Lignin substrates are highly crossed-linked aromatic heteropolymers, with complex and varied structures. However, relatively few genes in metagenomes have been predicted to be involved in lignin degradation, as exemplified by 75 gene families from the Amazon River microbiome and 13 gene families from North American forest microbiome participating in lignin degradation (8, 9).”

31) L578. *It is a bit lost here to read something relevant to "Amazon River". Better reorganization of this paragraph is expected.*

Response: Yes. We have revised the paragraph. The text now reads (*see Page 26, Line 620-625*):

“Cellulose and lignin degradation are mostly decoupled in aquatic ecosystems. For instance, cellulose and lignin degradation occurred in the different sections of Amazon River (9). In contrast, our *in situ* enrichment experiment not only suggested that lignin and hemi-/cellulose were co-degraded, but also revealed higher lignin degradation rates under lignocellulose substrates than that of purified lignin substrate (Fig. 1; also, see Table S2 at <https://doi.org/10.5281/zenodo.7659223>).”

32) L605. *Rewrite as "zones to link terrestrial and marine ecosystems"?*

Response: Yes. We have revised it accordingly. The text now reads (*see Page 27, Line 649-650*):

“In conclusion, coastal intertidal wetlands are important transitional zones to link terrestrial and marine ecosystems.”

33) L644-647. *I am not sure so much DNA could be yield with 20 mg samples. Better double check.*

Response: We have corrected it. The text now reads (*see Page 28, Line 687-690*):

“Approximately 20 g of each sample was collected for DNA extraction. DNA was extracted by the CTAB extraction method as previously described (10). As result, ~7-117 µg DNA was generated and preserved at -80°C for subsequent 16S rRNA amplicon sequencing and shotgun metagenomic sequencing.”

34) L648-655. *Provide proper references.*

Response: Yes. We have cited the references accordingly. The text now reads (*see Page 29, Line 691-699*):

“To investigate the enriched bacterial taxonomic compositions, all DNA samples (n=96) were used for 16S rRNA amplicon sequencing (Fig. 1A). The V4 region of the bacterial 16S rRNA gene with unique barcodes was amplified using the primer set 515F/806R (515F: 5'-GTGCCAGCMGCCGCGGTAA-3' and 806R: 5'-GGACTACHVGGGTWTCTAAT-3') (11). The amplified products were pooled in equimolar concentrations, and loaded onto an Illumina PE250 platform at Novogene Co., Ltd., Beijing, China, for high-throughput sequencing (12). A total of 6,259,640 reads were collected, with an average of 65,205 reads per sample and a standard deviation of 10,058.”

35) L656-664. *More information or reasonings are needed for inclusion of another experiments.*

Response: Yes. We have revised this paragraph to improve the readability. Please refer to our response to Review-Comment No. 10 above.

36) L695. *Provide proper references.*

Response: Yes. We have cited the literatures accordingly. The text now reads (*see Page 31, Line 742-745*):

“Similarly, representative genes, involved in lignin degradation, were annotated by searching against eggNOG (13), arCOG (14), KEGG (15) and COG (16) database with identity $\geq 70\%$ and e-value $\leq 1 e^{-5}$ (17, 18).”

37) L711. *Should be richness, evenness.*

Response: We have revised the statement. The text now reads (*see Page 31, Line 759-761*):

“For ASV community compositions, several indices of alpha diversity were calculated, including Shannon, Richness and Evenness. Comparisons of the alpha diversity between two groups was performed with Wilcoxon rank sum test.”

38) L720. *Please check whether Spearman or Pearson analysis were used in the Results section, and proper results should be reported.*

Response: Yes. We have revised it to improve the readability. Below are some of the examples.

(*see Page 9, Line 199-202*): “To discern the respective contributions of different taxa to the meta-community temporal pattern, the time-dissimilarity relationships of the top twenty most abundant taxa were examined via Spearman’s rank correlation analysis (see Fig. S3 at <https://doi.org/10.5281/zenodo.7659223>)”

(*see Page 10, Line 218-220*): “Mantel test further suggested substrate compositions (cellulose/hemicellulose/lignin) showed significant correlation with the community variance ($r = 0.25, p = 0.001$).”

39) L726. *More relevant references should be provided.*

Response: Yes. We have cited the literatures accordingly. The text now reads (*see Page 32-33, Line 774-793*):

“Null model

Phylogenetic bin-based null model analysis (iCAMP), with 16S rRNA gene sequencing data, was applied to quantify the contribution of various ecological processes to community assembly (19). It describes assembly processes, e.g., heterogeneous/ homogeneous selection, dispersal limitation, homogenizing dispersal, and drift, based on a quantitative framework (19). Briefly, the observed taxa were first divided into different phylogenetic bins based on their phylogenetic relationships. The process governing each bin was then identified based on null model analysis of phylogenetic diversity using beta Net Relatedness Index (β NRI) and taxonomic β -diversities using modified Raup-Crick metric (RC). For each bin, the fraction of pairwise comparisons with β NRI < -1.96 and $> +1.96$ were considered as the percentages of homogeneous and heterogeneous selection, respectively (20). Next, RC was used to partition the remaining pairwise comparisons with $|\beta$ NRI ≤ 1.96 : The fraction of pairwise comparisons with RC < -0.95 and $> +0.95$ are treated as the percentages of homogenizing dispersal and dispersal limitation (21), and the remaining fraction, with $|\beta$ NRI ≤ 1.96 and $|RC| \leq 0.95$, represented the percentages of drift (21). The above analysis was repeated for every bin, and then the fractions of individual processes across all bins were further weighted by the relative abundance of each bin and summarized to estimate the relative importance of individual processes at the whole community level (22).”

40) L758-817. *Please check the results relevant to qPCR and NMR are well reported in the Result section.*

Response: Yes. We have presented the results in the “Results” section.

For qPCR analysis, the text now reads (*see Page 19, Line 442-444*): “Hence, higher 16S rRNA copy number under lignocellulose substrates was observed, compared to that under lignin substrate (see Fig. S1B at <https://doi.org/10.5281/zenodo.7659223>).”

For NMR analysis, the text now reads (*see Page 14, Line 321-326*): “Consistent with the contents of guaiacyl (G-), syringyl (S-) and *p*-hydroxyphenyl (H-) types lignin units in all substrates (see Table S1 at <https://doi.org/10.5281/zenodo.7659223>), genes involved in H-type lignin units showed the greatest abundance, followed by genes taking parting in G-type lignin degradation and S-type lignin degradation, respectively (see Fig. S7A at <https://doi.org/10.5281/zenodo.7659223>).”

41) L768. *What kind of filter paper?*

Response: We have revised it to ensure understanding. The text now reads (*see Page 34, Line 818-819*):

“Approximately 8 g of each sample was mechanical disrupted, melt agar and then filtered by non-woven fabrics (30 g/m², 30 x 30 cm).”

References cited in this response-to-editor letter:

1. Xiao K, Liang X, Lu H, Li X, Zhang Z, Lu X, Wang H, Meng Y, Roy A, Luo W, Shen X, Irwin DM, Shen Y. 2022. Adaptation of gut microbiome and host metabolic systems to lignocellulosic degradation in bamboo rats. *ISME J* 16:1980-1992.
2. Reverter A, Ballester M, Alexandre PA, Marmol-Sánchez E, Dalmau A, Quintanilla R, Ramayo-Caldas Y. 2021. A gene co-association network regulating gut microbial communities in a Duroc pig population. *Microbiome* 9:52.
3. Shannon P, Markiel A, Ozier O, Baliga NS, Wang JT, Ramage D, Amin N, Schwikowski B, Ideker T. 2003. Cytoscape: a software environment for integrated models of biomolecular interaction networks. *Genome Res* 13:2498-2504.
4. Wang X, Lin L, Zhou J. 2021. Links among extracellular enzymes, lignin degradation and cell growth establish the models to identify marine lignin-utilizing bacteria. *Environ Microbiol* 23:160-173.
5. Hu A, Choi M, Tanentzap AJ, Liu J, Jang KS, Lennon JT, Liu Y, Soininen J, Lu X, Zhang Y, Shen J, Wang J. 2022. Ecological networks of dissolved organic matter and microorganisms under global change. *Nat Commun* 13:3600.
6. Xu X, Zarecki R, Medina S, Ofaim S, Liu X, Chen C, Hu S, Brom D, Gat D, Porob S, Eizenberg H, Ronen Z, Jiang J, Freilich S. 2019. Modeling microbial communities from atrazine contaminated soils promotes the development of biostimulation solutions. *ISME J* 13:494-508.
7. Hu B, Wang M, Geng S, Wen L, Wu M, Nie Y, Tang YQ, Wu XL. 2020. Metabolic exchange with non-alkane-consuming *Pseudomonas stutzeri* SLG510A3-8 improves n-alkane biodegradation by the alkane degrader *Dietzia* sp. Strain DQ12-45-1b. *Appl Environ Microbiol* 86:e02931-19.
8. Wilhelm RC, Singh R, Eltis LD, Mohn WW. 2019. Bacterial contributions to delignification and lignocellulose degradation in forest soils with metagenomic and quantitative stable isotope probing. *ISME J* 13:413-429.
9. Santos-Júnior CD, Sarmiento H, de Miranda FP, Henrique-Silva F, Logares R. 2020. Uncovering the genomic potential of the Amazon River microbiome to degrade rainforest organic matter. *Microbiome* 8:151.
10. Zhou J, Bruns MA, Tiedje JM. 1996. DNA recovery from soils of diverse composition. *Appl Environ Microbiol* 62:316-322.
11. Wear EK, Wilbanks EG, Nelson CE, Carlson CA. 2018. Primer selection impacts specific population abundances but not community dynamics in a monthly time-series 16S rRNA gene amplicon analysis of coastal marine bacterioplankton. *Environ Microbiol* 20:2709-2726.
12. Yu J, Guo M, Jiang W, Yang M, Pang X. 2020. Assessment of the microbiome and potential aflatoxin associated with the medicinal herb *Platyclusus orientalis*.

- Front Microbiol 11:582679.
13. Huerta-Cepas J, Szklarczyk D, Heller D, Hernández-Plaza A, Forslund SK, Cook H, Mende DR, Letunic I, Rattei T, Jensen LJ, von Mering C, Bork P. 2019. eggNOG 5.0: a hierarchical, functionally and phylogenetically annotated orthology resource based on 5090 organisms and 2502 viruses. *Nucleic Acids Res* 47:D309-314.
 14. Makarova KS, Wolf YI, Koonin EV. 2015. Archaeal clusters of orthologous genes (arCOGs): an update and application for analysis of shared features between *Thermococcales*, *Methanococcales*, and *Methanobacteriales*. *Life (Basel)* 5:818-840.
 15. Kanehisa M, Sato Y, Kawashima M, Furumichi M, Tanabe M. 2016. KEGG as a reference resource for gene and protein annotation. *Nucleic Acids Res* 44:D457-462.
 16. Galperin MY, Wolf YI, Makarova KS, Vera Alvarez R, Landsman D, Koonin EV. 2021. COG database update: focus on microbial diversity, model organisms, and widespread pathogens. *Nucleic Acids Res* 49:D274-281.
 17. Buchfink B, Xie C, Huson DH. 2015. Fast and sensitive protein alignment using DIAMOND. *Nat Methods* 12:59-60.
 18. Shen L, Liu Y, Allen MA, Xu B, Wang N, Williams TJ, Wang F, Zhou Y, Liu Q, Cavicchioli R. 2021. Linking genomic and physiological characteristics of psychrophilic *Arthrobacter* to metagenomic data to explain global environmental distribution. *Microbiome* 9:136.
 19. Ning D, Yuan M, Wu L, Zhang Y, Guo X, Zhou X, Yang Y, Arkin AP, Firestone MK, Zhou J. 2020. A quantitative framework reveals ecological drivers of grassland microbial community assembly in response to warming. *Nat Commun* 11:4717.
 20. Stegen JC, Lin X, Fredrickson JK, Konopka AE. 2015. Estimating and mapping ecological processes influencing microbial community assembly. *Front Microbiol* 6:370.
 21. Stegen JC, Lin X, Fredrickson JK, Chen X, Kennedy DW, Murray CJ, Rockhold ML, Konopka A. 2013. Quantifying community assembly processes and identifying features that impose them. *ISME J* 7:2069-2079.
 22. Zhou J, Ning D. 2017. Stochastic community assembly: Does it matter in microbial ecology? *Microbiol Mol Biol Rev* 81:e00002-e00017.

In conclusion, we greatly appreciate the reviewers' very helpful comments, and hope that these changes and responses now adequately address the concerns raised. All changes have been tracked in the marked-up manuscript.

Best regards,

Lu LIN, Ph.D

Professor

Institute of Marine Science and Technology, Shandong University, Qingdao, 266237, China

Email: linlu2019@sdu.edu.cn

March 27, 2023

Prof. Lu Lin
Shandong University
Qingdao
China

Re: mSystems01283-22R1 (Unraveling the roles of coastal bacterial consortia in degradation of various lignocellulosic substrates)

Dear Prof. Lu Lin:

Thank you for submitting your manuscript to mSystems. We have completed our review and I am pleased to inform you that, in principle, we expect to accept it for publication in mSystems. However, acceptance will not be final until you have adequately addressed the reviewer comments.

It is my pleasure to accept this manuscript. There are some English grammatical edits that must also be completed, could you please update the grammar and resubmit? Happy to accept once this is complete.

Preparing Revision Guidelines

- Upload a compare copy of the grammatically updated manuscript (without figures) as a "Marked-Up Manuscript" file.
- Each figure must be uploaded as a separate file, and any multipanel figures must be assembled into one file.
- Manuscript: A .DOC version of the revised manuscript
- Figures: Editable, high-resolution, individual figure files are required at revision, TIFF or EPS files are preferred

Sincerely,

Zarath Summers

Editor, mSystems

Journals Department
Reviewer comments:

Reviewer #2 (Comments for the Author):

The authors have replied my concerns well, and I have not further comments.

May 12, 2023

Prof. Lu Lin
Shandong University
Qingdao
China

Re: mSystems01283-22R2 (Unraveling the roles of coastal bacterial consortia in degradation of various lignocellulosic substrates)

Dear Prof. Lu Lin:

Your manuscript has been accepted, and I am forwarding it to the ASM Journals Department for publication. For your reference, ASM Journals' address is given below. Before it can be scheduled for publication, your manuscript will be checked by the mSystems production staff to make sure that all elements meet the technical requirements for publication. They will contact you if anything needs to be revised before copyediting and production can begin. Otherwise, you will be notified when your proofs are ready to be viewed.

If you would like to submit a potential Featured Image, please email a file and a short legend to msystems@asmusa.org. Please note that we can only consider images that (i) the authors created or own and (ii) have not been previously published. By submitting, you agree that the image can be used under the same terms as the published article. File requirements: square dimensions (4" x 4"), 300 dpi resolution, RGB colorspace, TIF file format.

We recognize that the video files can become quite large, and so to avoid quality loss ASM suggests sending the video file via <https://www.wetransfer.com/>. When you have a final version of the video and the still ready to share, please send it to mSystems staff at msystems@asmusa.org.

Sincerely,

Zarath Summers
Editor, mSystems

Journals Department
E-mail: mSystems@asmusa.org